# 3DGEER: 3D Gaussian Rendering Made Exact and Efficient for Generic Cameras

**Zixun Huang, Cho-Ying Wu**\*, **Yuliang Guo**\*, **Xinyu Huang, Liu Ren**
Bosch Research North America & Bosch Center for AI
https://zixunh.github.io/3d-geer

## Abstract

3D Gaussian Splatting (3DGS) (Kerbl et al., 2023) achieves an appealing balance between rendering quality and efficiency, but relies on approximating 3D Gaussians as 2D projections—an assumption that degrades accuracy, especially under generic large field-of-view (FoV) cameras. Despite recent extensions, no prior work has simultaneously achieved both *projective exactness* and *real-time efficiency* for general cameras. We introduce 3DGEER, a geometrically exact and efficient Gaussian rendering framework. From first principles, we derive a closed-form expression for integrating Gaussian density along a ray, enabling precise forward rendering and differentiable optimization under arbitrary camera models. To retain efficiency, we propose the *Particle Bounding Frustum (PBF)*, which provides tight ray–Gaussian association without BVH traversal, and the *Bipolar Equiangular Projection (BEAP)*, which unifies FoV representations, accelerates association, and improves reconstruction quality. Experiments on both pinhole and fisheye datasets show that 3DGEER outperforms prior methods across all metrics, runs $5\times$ faster than existing projective exact ray-based baselines, and generalizes to wider FoVs unseen during training—establishing a new state of the art in real-time radiance field rendering.

## 1 Introduction

Initiated by 3D Gaussian Splatting (3DGS) (Kerbl et al., 2023), volumetric particle rendering methods formulated under splatting (Yu et al., 2024a; Yan et al., 2024; Lu et al., 2024; Kheradmand et al., 2024; Charatan et al., 2024; Mallick et al., 2024) have gained significant popularity due to their impressive visual fidelity and high rendering speed. The core efficiency stems from EWA splatting (Zwicker et al., 2001), which uses a local affine approximation to project a 3D Gaussian as a 2D Gaussian on the image plane. While effective in narrow field-of-view (FoV) scenarios, this approximation introduces substantial errors in wide-FoV settings, where nonlinear distortions cause the true projection to deviate significantly from a symmetric 2D Gaussian—especially in fisheye images commonly encountered in robotics and autonomous driving (Rashed et al., 2021; Courbon et al., 2007; Kumar et al., 2023; Zhang et al., 2016; Yogamani et al., 2019; 2024; Sekkat et al., 2022). Although recent works have attempted to enable differentiable rendering under specific fisheye camera models (Liao et al., 2024) or more general camera models (Huang et al., 2024b), these approaches remain fundamentally constrained by their reliance on 2D projective-space approximations and have yet to yield an exact closed-form solution.

Meanwhile, Celarek et al. (2025) questions the necessity of exact rendering by scaling up the number of Gaussians (up to 1M) to mitigate projective approximation in splatting. However, the results are limited to small object-centric scenes. In large-scale scene reconstructions, the required number of Gaussians varies significantly across objects and frustums, making scaling up to larger scenes prohibitively costly. Further, we show projective errors under generic distorted cameras are substantially larger, where simply scaling up the Gaussian counts cannot close the quality gap (see Fig. 5).

A parallel line of work tackles volumetric particle rendering directly in a ray-tracing/marching formulation (Yu et al., 2024b; Condor et al., 2025; Mai et al., 2025; Moënne-Loccoz et al., 2024; Talegaonkar et al., 2025; Gu et al., 2025). These approaches naturally support large-FoV camera models

---

\*Corresponding Author: {yuliang.guo2, cho-ying.wu}@us.bosch.com

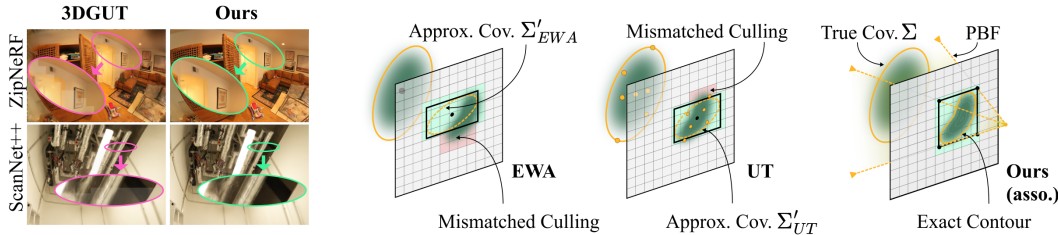

Figure 1: **Linear Approximation Error in Ray-Particle Association.** (*Left*) Grid-line artifacts caused by inaccurate UT association. (*Right*) Diagram illustrating our association and comparison with others. Our method avoids the intermediate conic approximation by directly computing the exact bounding structure from the true 3D covariance.

and are free from projective approximation error. However, achieving high efficiency remains a key challenge—particularly in associating each rendering ray with its most relevant 3D particles. To address this, significant effort has been devoted to accelerating it, such as adopting advanced Bounding Volume Hierarchies (BVH) employed in EVER (Mai et al., 2025) and 3DGRT (Moënne-Loccoz et al., 2024). Nevertheless, due to the inherent algorithmic complexity and difficulty in parallelization, pure ray-based methods have yet to match the efficiency of 3DGS. Alternatively, a hybrid approach has been proposed in 3DGUT (Wu et al., 2025) which adopts ray-tracing rendering with rasterization to solve ray-particle association efficiently. However, the inherited projective approximation error from the association stage still compromises rendering exactness in projective geometry and even risks grid-line artifacts (see Fig. 1 *Left*). To date, no existing method achieves both the geometric exactness of ray-based rendering and the high efficiency of splatting.

In this work, we propose 3DGEER—a novel 3D Gaussian Exact and Efficient Rendering method that achieves *exactness in projective geometry* (see definition in Sec. A) while maintaining *high efficiency comparable to splatting* for generic cameras (both perspective and a variety of fisheye projection models). 3DGEER is built upon three key contributions: (i) We revisit the heuristic maximum response function (Keselman & Hebert, 2022) of Gaussians from first principles and reveal its connection with the projective exactness. Based on it, we further derive the differentiable rendering framework with numerical stability. (ii) To perform efficient ray-particle association without compromising geometric exactness, we reformulate the problem as associating Camera Sub-frustums (CSF) with Particle Bounding Frustums (PBF)—analogous to the tile-AABB mapping used in 3DGS. For each 3D Gaussian, we derive a closed-form solution to compute the PBF tightly bounding it, enabling highly efficient association. (iii) In addition, we propose to uniformly sample rays in a novel Bipolar Equiangular Projection (BEAP) space to apply color supervision. This ray sampling strategy not only aligns image-space partitioning and the underlying CSF but also improves conventional pinhole or equidistant projections in reconstruction quality.

Extensive experiments on both fisheye and pinhole datasets—including ScanNet++ (Yeshwanth et al., 2023), Zip-NeRF (Barron et al., 2023), Aria (Lv et al., 2025), and MipNeRF (Barron et al., 2021)—show that 3DGEER consistently outperforms prior and concurrent methods, achieving 0.9–4.8 PSNR gains under challenging wide-FoV conditions, and 0.3–2.0 on pinhole views. Remarkably, our approach even surpasses splat-based methods on wide-FoV cameras when trained only with narrow-FoV data, highlighting its strong generalization to extremely wide-FoV rendering.

## 2 RELATED WORKS

### 2.1 SPLATTING-BASED GAUSSIAN RENDERING

Most 3DGS variants (Yu et al., 2024a; Zhang et al., 2024; Rota Bulò et al., 2024; Gao et al., 2025; Zhao et al., 2025; Kerbl et al., 2024; Hou et al., 2025; Liu et al., 2025) adopt the original EWA splatting strategy (Zwicker et al., 2001; Ren et al., 2002), projecting 3D Gaussians onto the image plane and approximating their appearance as 2D conics. To support differentiable rendering under large-FoV camera models with distortion, methods such as FisheyeGS (Liao et al., 2024) and GS++ (Huang et al., 2024b) estimate the Jacobian of equidistant or spherical projections to adjust the 2D mean and covariance of the projected Gaussians. However, these splatting-based approaches still rely on a first-order Taylor expansion of the highly non-linear projection function. In wide-FoV

Table 1: **Summary of Particle Rendering Methods w.r.t. Projective Exactness.** The top section lists splatting-based methods, while the bottom section presents ray-based approaches—some of which still rely on projective approximated association (e.g., EWA or Unscented Transform).

| Method | Render | Particle Asso. | | Particle Type | Large FoV | High FPS | Exact Meth. |
| | | Method | Scope | | | | |
|---|---|---|---|---|---|---|---|
| 3DGS (Kerbl et al., 2023) | Splat. | EWA | Img. | 3D Gauss. | ✗ | ✓ | ✗ |
| FisheyeGS (Liao et al., 2024) | Splat. | EWA | Img. | 3D Gauss. | ✓ | ✓ | ✗ |
| GS++ (Huang et al., 2024b) | Splat. | EWA | Img. | 3D Gauss. | ✗ | ✓ | ✗ |
| 2DGS (Huang et al., 2024a) | Ray | Splat Bound. | Img. | 2D Gauss. | ✗ | ✓ | ✓ |
| GOF (Yu et al., 2024b) | Ray | EWA | Img. | 3D Gauss. | ✗ | ✗ | ✗ |
| EVER (Mai et al., 2025) | Ray | BVH | Scene | 3D Ellip. | ✓ | ✗ | ✓ |
| 3DGRT (Moënne-Loccoz et al., 2024) | Ray | BVH | Scene | 3D Gauss. | ✓ | ✗ | ✓ |
| 3DGUT (Wu et al., 2025) | Ray | UT | Img. | 3D Gauss. | ✓ | ✓ | ✗ |
| HTGS (Hahlbohm et al., 2025) | Ray | Splat Bound. | Img. | 3D Gauss. | ✗ | ✓ | ✓ |
| 3DGEER (Ours) | Ray | PBF | Frust. | 3D Gauss. | ✓ | ✓ | ✓ |

settings, higher-order terms are non-negligible, resulting in significant approximation errors and degraded reconstruction quality.

Fundamentally, the true projection of a 3D Gaussian under non-linear camera models—whether ideal pinhole or distorted projections—is not a symmetric 2D Gaussian. In other words, any framework that relies on linear projective geometry to approximate inherently nonlinear transformations will inevitably introduce approximation errors that cannot be fully eliminated. In contrast, our approach avoids such projective approximations entirely by aggregating Gaussian density directly along rays in a volumetric rendering formulation.

## 2.2 RAY-BASED GAUSSIAN RENDERING

Keselman & Hebert (2022) introduces an assumption that a ray intersects the Gaussian density function at the maximum intensity response. Building on this idea, 3DGRT (Moënne-Loccoz et al., 2024) adopts this heuristic intersection to establish a 3D Gaussian ray-tracing framework, which enables ray-space sampling without explicit projection and alleviates wide-FoV distortions. However, 3DGRT introduces substantial computational overhead, as the ray–particle association relies on costly scene-level spatial partitioning structures for BVHs, ultimately resulting in low frame rates.

Alternatively, EVER (Mai et al., 2025) replaces the Gaussian representation with simplified ellipsoids of constant density, enabling a closed-form solution for transmittance-weighted ray integration. However, this substitution reduces the expressive capacity of the representation compared to full 3D Gaussians and still suffers from limited runtime efficiency due to its reliance on BVH-based structures. 2DGS (Huang et al., 2024a) substitutes 3D Gaussians with 2D surfels, allowing exact ray–surface intersections. HTGS (Hahlbohm et al., 2025) reintroduces the full Gaussian expressions while adopting 2DGS's ray–splat association. Yet, because its association remains restricted to screen space, it falls short of ensuring projective exactness under wider FoVs. Our contribution lies in deriving closed-form bounds and a full rendering pipeline directly in a camera-agnostic angular domain (see details in Sec. D) and we especially discuss the relation and differences in computation to prior 2DGS and HTGS parametrization and rendering in Sec. D.3.

3DGUT (Wu et al., 2025), a follow-up to 3DGRT, improves runtime efficiency by adopting splatting-based particle association, achieving frame rates comparable to 3DGS. However, its acceleration relies on the Unscented Transform (UT), a sampling-based approximation used to estimate 2D conics in projective image space. A similar limitation exists in GOF (Yu et al., 2024b), which directly applies the EWA trick to fit 2D bounding conics and projective centers for association. Notably, under extreme-FoV conditions, imprecise AABB estimation and the resulting inaccurate tile-to-AABB associations can lead to visible grid-line artifacts (see Fig. 1 *Left*).

In contrast to these methods, our approach solves the ray-particle association problem directly at the frustum level in 3D space (see Fig. 1 *Right*), preserving the geometric exactness of ray-based rendering across arbitrary camera models while maintaining rendering speeds comparable to 3DGS. Tab. 1 summarizes representative methods that aim to address projective approximation errors, highlighting key differences across several important dimensions.

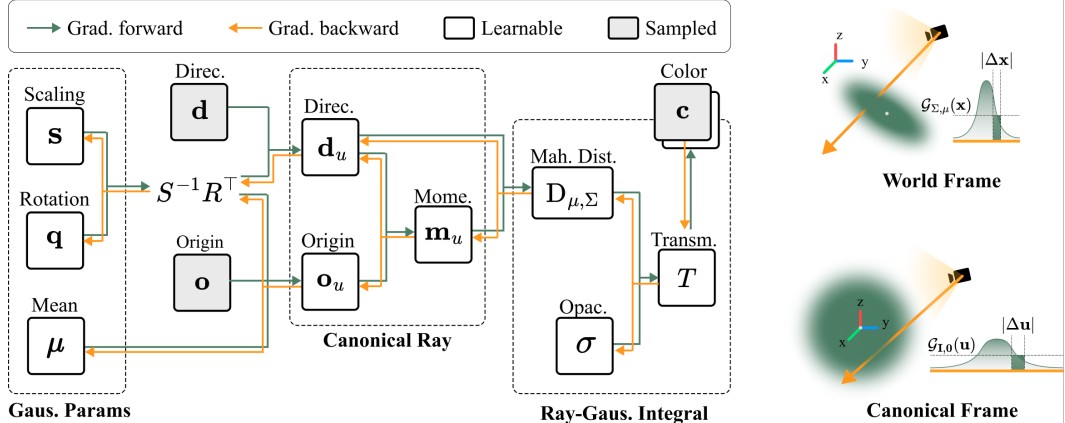

Figure 2: **Gradient Flow and Canonical Transformation.** (*Left*) Illustration of forward and backward gradient propagation from the Gaussian parameters s, q, $\boldsymbol{\mu}$, and the opacity coefficient $\sigma$ to the transmittance. (*Right*) Integral of 3D Gaussian density along a ray, where the green box highlights the product of density and ray segment length (see Eq. B.5).

## 3 METHOD

We propose 3DGEER, a Gaussian rendering framework that achieves both geometric exactness and real-time efficiency under arbitrary FoV cameras. Our method addresses the two key bottlenecks of prior work—*projection approximation* in splatting-based rendering and *BVH overhead* in ray-based rendering—through three complementary components:

- Ray-based rendering formulation (Sec. 3.1): a closed-form differentiable solution for integrating Gaussian density along 3D rays, eliminating linearization error from projection-based splatting.

- Particle Bounding Frustum (PBF) (Sec. 3.2): a novel ray–particle association scheme that replaces BVH traversal with exact frustum–particle tests, yielding both accuracy and efficiency.

- Bipolar Equiangular Projection (BEAP) (Sec. 3.3): an image representation unifying arbitrary FoV camera models, which accelerates PBF computation and improves reconstruction quality.

The Appendix further provides: (i) the necessary preliminaries and notations (Sec. B.1), (ii) the complete proof to Sec. 3.1 establishing the connection between maximum response and projective exactness (Sec. B), (iii) the full derivations of backward gradients (Sec. C) and its numerical stability, and (iv) the full derivations of geometrically exact association (Sec. D).

### 3.1 RENDERING WITH EXACT PROJECTIVE GEOMETRY

One key challenge in geometrically exact Gaussian rendering is to obtain a closed-form expression for integrating density and color along a ray. Our strategy is to map each anisotropic Gaussian in world space into a canonical coordinate system where all Gaussians become isotropic. This reduces the transmittance integral to a measure-preserving change of variables, admitting a simple closed-form solution.

**Canonical transformation.** For each Gaussian with mean $\boldsymbol{\mu}$, rotation $R$, and scaling $S$, we define the mapping

$$\begin{bmatrix} \mathbf{x} \\ 1 \end{bmatrix} = \begin{bmatrix} RS & \boldsymbol{\mu} \\ \mathbf{0} & 1 \end{bmatrix} \begin{bmatrix} \mathbf{u} \\ 1 \end{bmatrix}, \tag{1}$$

which maps a canonical coordinate $\mathbf{u}$ to world coordinates $\mathbf{x}$. This implies $\Delta\mathbf{x} = RS\Delta\mathbf{u}$ and absorbs the Jacobian determinant into the density measure (see Fig. 2 *Right*). Consequently, each Gaussian reduces to the shared isotropic form

$$\mathcal{G}_{\mathbf{I},\mathbf{0}}(\mathbf{u}) = \frac{1}{\rho} \exp\left(-\tfrac{1}{2}\|\mathbf{u}\|^2\right). \tag{2}$$

**Closed-form transmittance.** For a ray $\mathbf{r}(t) = \mathbf{o} + t\mathbf{d}$, its canonical form is $\mathbf{r}_u(t) = \mathbf{o}_u + t\mathbf{d}_u$ with

$$\mathbf{o}_u = S^{-1}R^\top(\mathbf{o} - \boldsymbol{\mu}), \quad \mathbf{d}_u = S^{-1}R^\top\mathbf{d}. \tag{3}$$

The accumulated transmittance is then

$$T(\mathbf{o}, \mathbf{d}) = \sigma \int_{t\in\mathbb{R}} \mathcal{G}_{\mathbf{I},\mathbf{0}}(\mathbf{o}_u + t\mathbf{d}_u)\, dt = \sigma \exp\left(-\tfrac{1}{2}D_{\mu,\Sigma}^2(\mathbf{o}, \mathbf{d})\right), \tag{4}$$

where

$$D_{\mu,\Sigma}^2(\mathbf{o}, \mathbf{d}) = \frac{\|\mathbf{o}_u \times \mathbf{d}_u\|^2}{\|\mathbf{d}_u\|^2} \tag{5}$$

is the squared perpendicular (Mahalanobis) distance from the ray to the Gaussian center.

**Discussion.** This result reveals that the transmittance depends only on the ray's distance to the Gaussian in the canonical space, yielding an efficient and geometrically exact closed-form solution. Interestingly, this expression is equivalent to the "maximum response" heuristic used in prior works (Keselman & Hebert, 2022; Moënne-Loccoz et al., 2024), while our mathematical derivation reveals its underlying projective exactness (see details in Sec. B). Furthermore, by following our closed-form derivation process and the corresponding backward gradient chain (see Fig. 2 *Left* and details in Sec. C), we explicitly avoid the numerical instabilities that prior ray-based work (Yu et al., 2024b) encounters and eliminates the need for redundant post-processing, such as filtering out degenerate Gaussians.

### 3.2 RAY-PARTICLE ASSOCIATION WITH EXACT PROJECTIVE GEOMETRY

Given a ray-based Gaussian rendering method, both its final accuracy—particularly under large-FoV camera models—and its runtime efficiency depend heavily on the ray-particle association strategy, which has been largely overlooked in prior works. To determine which Gaussians contribute to each ray, we simplify the association problem by mapping Camera Sub-Frustums (CSFs) to Particle Bounding Frustums (PBFs)—the tight frustums enclosing each Gaussian at a certain distance from its center (Fig. 3). A Gaussian is assigned to a CSF if its PBF intersects the CSF. All Gaussians associated with a CSF are then considered contributors to every ray within that CSF.

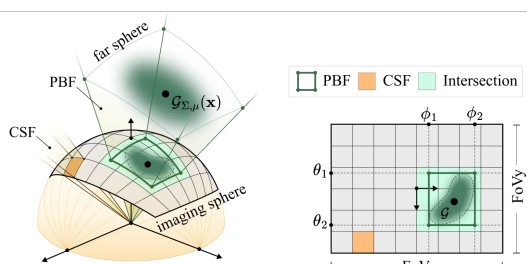

Figure 3: **PBF-CSF Association.** (*Left*) PBF defined by four planes tangent to a 3D Gaussian ellipsoid. (*Right*) The intersection between the PBF and CSF, unfolded onto the BEAP imaging plane.

To efficiently compute a PBF, we first define two spherical angles (see Fig. E.1) in the camera space:

$$\theta = \arctan\left(\frac{\mathbf{d}_{c,x}}{\mathbf{d}_{c,z}}\right), \quad \phi = \arctan\left(\frac{\mathbf{d}_{c,y}}{\mathbf{d}_{c,z}}\right), \tag{6}$$

where $(\mathbf{d}_{c,x}, \mathbf{d}_{c,y}, \mathbf{d}_{c,z})$ denotes the normalized direction vector pointing to the camera (view) space point $\mathbf{d_c}$. The angles $\theta$ and $\phi$ correspond to the ray's incidence angles in the horizontal $(xz)$ and vertical $(yz)$ planes, respectively, each ranging from $-\frac{\pi}{2}$ to $\frac{\pi}{2}$. We define a PBF by four planes — two aligned with the $\theta$ angle and two with the $\phi$ angle (see Fig. 3):

$$\mathbf{g}_{\theta_{1,2}} = (-1, 0, \tan\theta_{1,2}, 0)^\top, \quad \mathbf{g}_{\phi_{1,2}} = (0, -1, \tan\phi_{1,2}, 0)^\top, \tag{7}$$

where $\theta_1 < \theta < \theta_2$ and $\phi_1 < \phi < \phi_2$ specify the angular limits.

Given a camera with extrinsic matrix $V = [R_c \mid \mathbf{t}_c] \in \mathrm{SE}(3)$ and a canonical-to-world transformation matrix $H$ (See Eq. B.4), we apply the inverse transpose of the camera-to-canonical matrix $(VH)^{-1}$ to parameterize the corresponding canonical planes using the angular limits defined in view space. This transformation enables us to effectively derive the constraints for the PBFs. Take the bounds of $\theta$ as an example, the canonical plane is given by:

$$\mathbf{g}_{u_{1,2}} = (VH)^\top \mathbf{g}_{\theta_{1,2}} = -(VH)_0 + \tan\theta_{1,2}(VH)_2, \tag{8}$$

where $(VH)_i$ denotes the $i$-th row of the matrix, expressed as a column vector.

Assuming a Gaussian ellipsoid defined by a $\lambda$-standard-deviation contour, we compute the angular bounds of the local frustum by solving the following quadratic equations, which enforce that the transformed planes are tangent to the ellipsoidal contour:

$$\left((\lambda^2, \lambda^2, \lambda^2, -1) \circ \mathbf{g}_u^\top\right) \cdot \mathbf{g}_u = 0, \tag{9}$$

where $\circ$ denotes the Hadamard (element-wise) product, and $\lambda$ is a parameter that controls the spread of the contour, typically set to 3. An identical constraint is applied to $\mathbf{g}_v^\top$ in order to compute the corresponding angular bounds $\phi_{1,2}$. The roots of these quadratic equations determine the tangent values of the angular bounds $[\theta_1, \theta_2]$ and $[\phi_1, \phi_2]$, which define the PBF angular limits. Notably, solving these quadratic equations under canonical space constraints respects the true 3D covariance $\Sigma_c$ and 3D mean $\boldsymbol{\mu}_c$ of the Gaussian in the camera space. Specifically, for the bounds of $\theta$, we have $c = \tan \theta_{1,2}$ satisfying:

$$\mathcal{T}_{22}c^2 - 2\mathcal{T}_{02}c + \mathcal{T}_{00} = 0, \quad \text{where} \quad \mathcal{T}_{ij} = \lambda^2 \Sigma_c^{i,j} - \left(\boldsymbol{\mu}_c \boldsymbol{\mu}_c^\top\right)^{i,j}. \tag{10}$$

This formulation is efficient to solve (See details in Sec. D.1) and eliminates the need for intermediate conic approximations, as in EWA or UT, as well as the screen-space constraints in HTGS and 2DGS that limit their FoV.

### 3.3 BIPOLAR EQUIANGULAR PROJECTION (BEAP)

We introduce a novel BEAP image representation for several reasons. First, it effectively represents images from large-FoV cameras without introducing FoV loss. Second, it allows image tiles and their corresponding CSFs to share an identical parameterization in terms of two $\theta$ and two $\phi$ angles, enabling highly efficient PBF association and optimal GPU parallelization. Third, the resulting ray sampling achieves a more uniform distribution in the 3D space compared to conventional projective rendering, which empirically improves training efficiency and reconstruction quality.

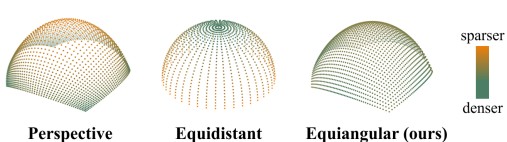

Figure 4: **Comparison of Ray Distributions under Varying Projections.** We project pixels from three imaging spaces onto the unit sphere to compare ray distributions. Among them, our BEAP projection achieves the most uniform coverage.

Specifically, we evenly sample the rays using the spherical angular coordinates $(\theta, \phi)$ (See Eq. 6), and then map the rays back to a discrete w $\times$ h image space using a linear transformation consisting of a scaling factor and a center shift as:

$$(x, y) = \left\lfloor \frac{\mathrm{w}\theta}{\mathrm{FoV}_x} + \frac{\mathrm{w}+1}{2}, \frac{\mathrm{h}\phi}{\mathrm{FoV}_y} + \frac{\mathrm{h}+1}{2} \right\rfloor, \tag{11}$$

where $\mathrm{FoV}_x, \mathrm{FoV}_y$ indicate the camera's horizontal and vertical field of view. As illustrated in Fig. 4, sampling rays uniformly in $(\theta, \phi)$ leads to more balanced spatial coverage within the view frustum. Our sampling strategy contrasts with projective sampling, which disproportionately allocates samples toward peripheral regions, and equidistant projection, which oversamples near the image center.

## 4 EXPERIMENTS

In this section, we evaluate our proposed method, 3DGEER, across multiple datasets featuring both pinhole and fisheye camera models. All experiments are conducted using an RTX 4090 GPU and 64GB of RAM as the default configuration, unless otherwise stated. Full implementation details are provided in the Sec. G.

**Datasets.** We evaluate performance on generic distorted cameras, where projective errors are critical, on all scenes from Fisheye ZipNeRF (Barron et al., 2023), five from ScanNet++ (Yeshwanth et al., 2023), and two from Aria (Lv et al., 2025). ZipNeRF/ScanNet++ cameras have 180° diagonal FoV, and Aria has 110° circular FoV. For pinhole settings, we use Pinhole ZipNeRF and MipNeRF360 (Barron et al., 2021).

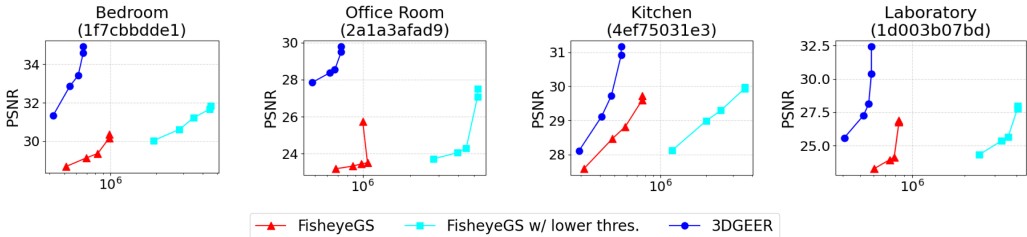

Figure 5: **PSNR↑ Trends When Scaling-Up Gaussian Counts.** Brute-force scaling fails to close the gap in projective exactness (see full metrics in Fig. K.3 and Tab. K.3).

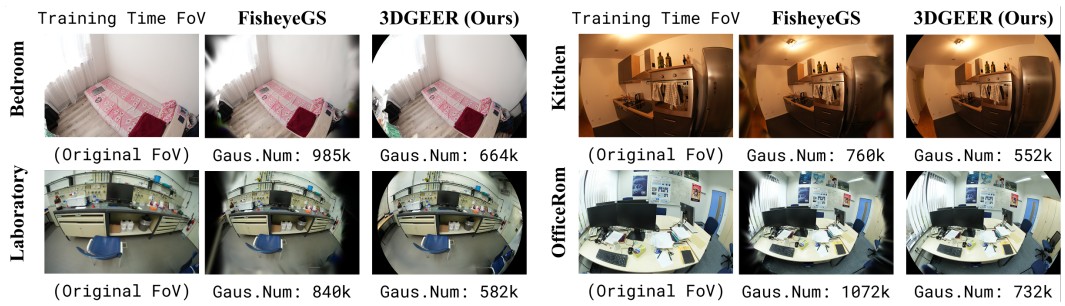

Figure 6: **Extreme-FoV Generalization.** 3DGEER reconstructs complete scenes with fewer Gaussians, while FisheyeGS suffers from severe quality degradation.

**Evaluation Metrics and Protocol.** We report PSNR↑, SSIM↑, LPIPS↓, runtime performance (FPS↑) and Gaussian counts↓. To avoid representation bias, all results are projected back to the image space for full-FoV evaluation after 30k iterations.

### 4.1 Scaling Gaussians vs. Projective Exactness in Wide-FoV Cameras

(Celarek et al., 2025) argues that simply scaling the number of Gaussians (up to 1M) suffices to mitigate the linear approximation gap between splat-based and ray-based methods. However, this strategy has only been demonstrated on small object-centric scenes and does not address the challenges posed by wide-FoV distortions. To further investigate, we scale up the splat-based fisheye method (FisheyeGS) on ScanNet++ scenes by lowering its densification threshold, producing 3.6–5M Gaussians (H200).

As shown in Fig. 5, the red curve (FisheyeGS) and blue curve (3DGEER) show a clear PSNR gap under the same Gaussian count, implying an approximation gap in projective exactness (from 4k to 30k iterations, after which densification typically saturates). The light-blue curve further shows the scaled-up FisheyeGS, despite its substantial increase in memory and computation, performance saturates at 29.3 PSNR, whereas ours achieves 32.1 with only 550-700K Gaussians. Full statistics are reported in Tab. K.3.

### 4.2 Comprehensive Evaluation on All-FoV Novel View Synthesis

We first evaluate 3DGEER against representative baselines on both fisheye and pinhole benchmarks. To ensure comprehensive evaluations, we incorporate additional analysis tailored to each dataset.

**ScanNet++.** As shown in Tab. 2, 3DGEER consistently outperforms all baselines across metrics. To examine distortion effects, we separately evaluate central and peripheral regions (details in Sec. G). Competing methods perform reasonably at the center but degrade at the periphery, whereas 3DGEER maintains high fidelity throughout, producing sharper boundaries and fewer artifacts (Fig. K.4). Notably, even when trained only on central crops, 3DGEER still surpasses FisheyeGS across all regions.

Table 2: **Comparison on the ScanNet++ Dataset.** Each method is trained either on the full FoV or only the central region, and then evaluated on the full, central, and peripheral regions to assess robustness under varying distortions. Region-wise metrics are weighted by pixel masks, while LPIPS is excluded since it is not directly separable by region (Tab. K.2 shows full stats).

| Method | Training FoV | Training Space | Full FoV PSNR ↑ | SSIM ↑ | LPIPS ↓ | Central PSNR ↑ | SSIM ↑ | Peripheral PSNR ↑ | SSIM ↑ |
|---|---|---|---|---|---|---|---|---|---|
| 3DGS | Central | Persp. | - | - | - | 31.26 | 0.948 | - | - |
| FisheyeGS | Full | Eq. | 27.81 | 0.946 | 0.139 | 32.44 | 0.956 | 23.28 | 0.914 |
| EVER | Full | KB | 29.47 | 0.924 | 0.167 | 29.93 | 0.924 | 28.72 | 0.925 |
| 3DGUT | Full | Eq. | 30.64 | 0.944 | 0.150 | 31.87 | 0.945 | 28.84 | 0.937 |
| 3DGEER (Ours) | Central | BEAP | 29.93 | 0.949 | 0.130 | **32.86** | 0.954 | 26.64 | 0.930 |
| 3DGEER (Ours) | Full | BEAP | **31.50** | **0.953** | **0.126** | 32.64 | 0.955 | **28.94** | **0.945** |

Table 3: **Comparison on the ZipNeRF Dataset.** The experiments follow a cross-camera generalization setup, where each method is trained on either Fisheye (FE, 1/8 resolution) or Pinhole (PH, 1/4 resolution) data and evaluated separately on both Fisheye and Pinhole test sets (Tab. K.9 shows full stats).

| Train Data Test Data Method | 1/8 FE FE PH PSNR ↑ | 1/8 FE FE PH SSIM ↑ | 1/8 FE FE PH LPIPS ↓ | 1/4 PH FE PH PSNR ↑ | 1/4 PH FE PH SSIM ↑ | 1/4 PH FE PH LPIPS ↓ |
|---|---|---|---|---|---|---|
| FisheyeGS | 23.18  26.44 | 0.858  0.868 | 0.211  0.239 | 19.43  26.61 | 0.791  **0.889** | 0.247  **0.212** |
| EVER | 25.17  26.14 | 0.880  0.851 | 0.153  0.237 | 22.90  26.45 | 0.835  0.862 | **0.207**  0.222 |
| 3DGUT | 24.77  25.59 | 0.879  0.804 | 0.183  0.324 | 18.61  25.96 | 0.662  0.841 | 0.300  0.270 |
| 3DGEER (Ours) | **26.24**  **27.62** | **0.897**  **0.888** | **0.140**  **0.214** | **23.39**  **27.61** | **0.846**  0.863 | 0.209  0.254 |

Qualitatively (Fig. 6), we evaluate extreme-FoV generalization by training on the original dataset FoV (180° diagonal FoV) but testing at much wider FoVs (180° FoV, circular shape). 3DGEER reconstructs complete scenes with fewer Gaussians, while splatting-based methods such as FisheyeGS degrade significantly at the periphery (see more visuals in Fig. K.1-K.2 with extreme FoV).

**ZipNeRF**. The dataset provides frame-wise aligned pinhole and fisheye views. Beyond standard fisheye evaluation, we conduct *cross-camera generalization*—training on fisheye and testing on pinhole, and vice versa—to assess robustness. As shown in Tab. 3, 3DGEER consistently outperforms all methods across metrics, fully exploiting large-FoV training data via geometrically exact rendering. Trained on pinhole, 3DGEER still yields notable PSNR gains on both views. In contrast, FisheyeGS performs well only on pinhole-to-pinhole, and 3DGUT drops sharply when tested on fisheye after pinhole training. Overall, EVER and 3DGEER generalize better due to geo-

Table 4: **Quantitative Results on the MipNeRF360 Dataset.** In addition to quality metrics, FPS is reported using an RTX 4090 as the default benchmark. Numbers sourced from original works are marked with [†] (measured on RTX 6000 Ada) and [‡] (measured on RTX 5090).

| Method | PSNR ↑ | SSIM ↑ | LPIPS ↓ | FPS↑ |
|---|---|---|---|---|
| 3DGS | 27.21 | 0.815 | 0.214 | **343** |
| EVER | 27.51 | **0.825** | 0.233 | 36 |
| 3DGRT | 27.20 | 0.818 | 0.248 | 52[†] / 68[‡] |
| 3DGUT | 27.26 | 0.810 | 0.218 | 265[†] / 317[‡] |
| 3DGEER (Ours) | **27.76** | 0.821 | **0.210** | 327 |

metric exactness, while 3DGEER is superior in all cases. Qualitatively (Fig. 7), 3DGEER recovers finer details and shows stronger resistance to artifacts in the periphery in the most challenging pinhole-to-fisheye setting.

We further evaluate 3DGEER on the **MipNeRF360** dataset under standard narrow-FoV conditions. As shown in Tab. 4, 3DGEER sets the new state-of-the-art across all the metrics. It is over $5\times$ faster than exact ray-marching/tracing methods such as EVER and 3DGRT, and the only projective-exact method with runtime comparable to 3DGS. Moreover, it consistently surpasses projective-approximation baselines (3DGS, 3DGUT) in PSNR, SSIM, and LPIPS. Additional quantitative results on **Aria**, an egocentric fisheye dataset (moderate 110° FoV, circular shape) with low-light imaging, and more visuals are provided in the Sec. J.

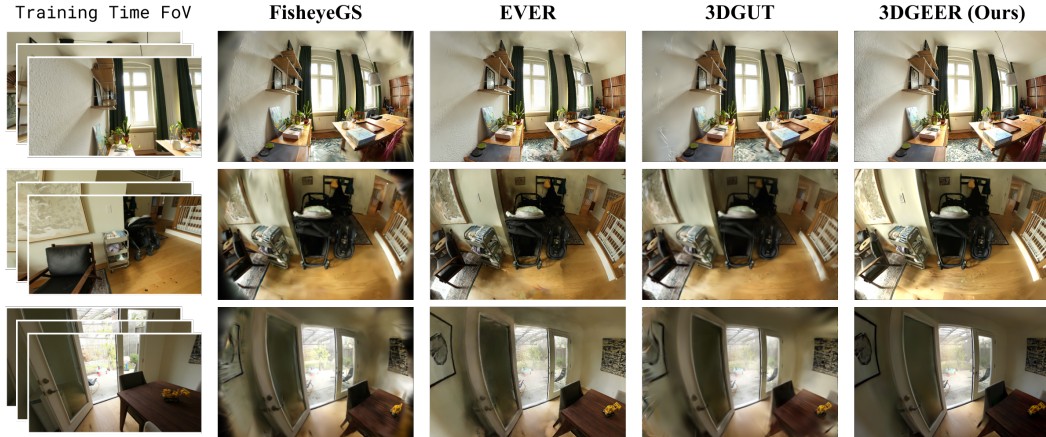

| Training Time FoV | FisheyeGS | EVER | 3DGUT | 3DGEER (Ours) |

Figure 7: **Qualitative Results on the ZipNeRF Dataset with Cross-Camera Generalization.** Performance differences are pronounced in the most challenging Pinhole (PH) train – Fisheye (FE) test setup, where our method demonstrates a significant advantage.

Table 5: **Training-Time Transmittance Replacement on ScanNet++.** Using splatting-based transmittance increases the Gaussian count due to larger approximation error, yet the perceptual gap remains compared with our projective-exact formulation (Tab. K.7 shows the full stats).

| Transmittance Method | Full FoV | | | Central | | Peripheral | | Avg. Gaussian Number (k) ↓ |
|---|---|---|---|---|---|---|---|---|
| | PSNR ↑ | SSIM ↑ | LPIPS ↓ | PSNR ↑ | SSIM ↑ | PSNR ↑ | SSIM ↑ | |
| 3DGS Splats | 22.86 | 0.878 | 0.177 | 27.16 | 0.884 | 18.56 | 0.855 | 1396.4 |
| FisheyeGS Splats | 27.90 | 0.948 | 0.141 | 32.49 | 0.957 | 23.36 | 0.917 | 920.6 |
| Projective-Exact (Ours) | **31.50** | **0.953** | **0.126** | **32.64** | **0.955** | **28.94** | **0.945** | **591.8** |

## 4.3 COMPREHENSIVE RUNTIME ANALYSIS

We further analyze runtime on RTX 4090 by comparing 3DGEER with splatting-based methods (3DGS on **MipNeRF360** and FisheyeGS on **ScanNet++**). Although our ray-based rendering incurs slightly higher cost in the rendering stage, PBF's efficient ray-particle association greatly reduces Gaussian duplication and sorting overhead (See detailed runtime in Tab. K.4), resulting in overall efficiency comparable to optimized splatting pipelines.

Specifically, PBF yields very tight bounds, with each tile associated with only ∼475 Gaussians on average—3–5× fewer than EWA or UT (Tab. K.6). This directly explains the 2.5–5× runtime speedup in the association stage (Tab. K.5). In contrast, projection-based schemes (EWA, UT) conservatively overestimate screen-space extent, leading to excessive overdraw and inefficient memory access. Even when accelerated by SnugBox (Hanson et al., 2025), they remain constrained by mis-centered density approximations, often exacerbating artifacts (Fig. 1). Beyond speed, PBF preserves geometric accuracy and numerical stability, eliminating projective approximation errors. As shown in Tab. K.6, even when ray-based rendering Gaussian fields trained under other association schemes, PBF remains artifact-free.

## 4.4 LOCAL AFFINE APPROXIMATION VS. PROJECTIVE-EXACT INTEGRAL

To isolate the effect of our projective-exact transmittance formulation (Eq. 4), we replace it with the commonly used local affine splats (i.e., 2D conic as approximated in 3DGS and FisheyeGS).

**Rendering-Time Approx.** To better highlight the difference between the two rendering formulations, we replace the transmittance with the splatting-based approximation only at test time, while keeping all projective-exact components during training. As shown in Fig. 8, when the transmittance relies on the splat approximation but the associations are computed using our exact projective geometry, their inconsistency leads to mismatched culling. This experiment magnifies and explains the gridline artifact reported in 3DGUT (Fig. 1), where the mismatch arises in the opposite direction—ray-based transmittance combined with conic-approximate association.

| 3DGS Splats + PBF | FisheyeGS Splats + PBF | 3DGEER (Ours) | GT |

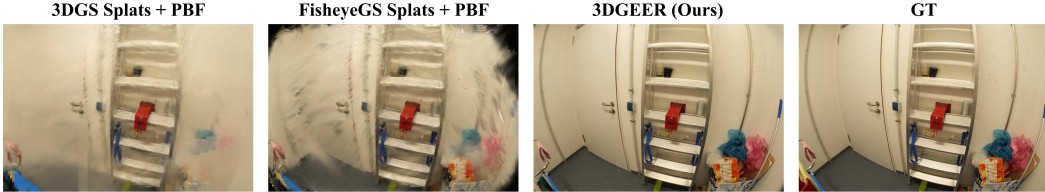

Figure 8: **Rendering-Time Transmittance Func. Replacement.** Replacing projective-exact transmittance with the splatting approximation only at test time leads to mismatched culling.

Table 6: **BEAP vs. Alternative Supervision Schemes.** BEAP consistently outperforms other projections across all metrics on ScanNet++. (Tab. K.8 shows full stats).

| Method | Training FoV | Training Space | Full FoV | | | Central | | Peripheral | |
|---|---|---|---|---|---|---|---|---|---|
| | | | PSNR ↑ | SSIM ↑ | LPIPS ↓ | PSNR ↑ | SSIM ↑ | PSNR ↑ | SSIM ↑ |
| 3DGEER w/o BEAP | Central | Persp. | 29.84 | 0.943 | 0.131 | 32.21 | 0.951 | 26.23 | 0.915 |
| 3DGEER w/o BEAP | Full | Persp. | 21.11 | 0.853 | 0.300 | 21.32 | 0.855 | 20.46 | 0.846 |
| 3DGEER w/o BEAP | Full | Eq. | 31.05 | 0.948 | 0.135 | 32.21 | 0.952 | 28.56 | 0.935 |
| 3DGEER (Ours) | Full | BEAP | **31.50** | **0.953** | **0.126** | **32.64** | **0.955** | **28.94** | **0.945** |

**Training-Time Approx.** As shown in Tab. 5 experimented on **ScanNet++**, when splatting-based transmittance is also used during training while association remains projective-exact, the optimizer and densification partially compensate for the approximation error and the cross-view inconsistency introduced by splatting. The larger loss induces stronger gradients throughout training, ultimately leading to a larger (1.6–2.4×) number of Gaussians being added to fit the scene. However, despite this increased Gaussian count, the perceptual gap introduced by the approximation remains, indicating that the error cannot be fully corrected through densification alone.

### 4.5 BEAP On/Off Study

We further compare BEAP supervision against other widely used projection models on **ScanNet++**. In this experiment, we keep the projective-exact integral and PBF for Gaussian transmittance and association fixed, and vary only the ray-sampling distribution used for supervision (pinhole, equidistant, and BEAP). As shown in Tab. 6 and Fig. K.5, our BEAP-based supervision outperforms all alternatives across every metric. Its more uniform sampling provides smoother and more consistent gradients, enabling better convergence on fine details.

The discrepancy becomes more pronounced under pinhole cameras. Under a fixed resolution, a well-balanced information distribution can matter more than simply using a larger FoV. Notably, when training in the perspective domain, the full FoV can even underperform the central region (see Tab. 6, rows 1–2) because large-FoV images (Fig. K.2, *Left*) have highly non-uniform angular sampling: informative content is concentrated at the center, while the less informative periphery occupies most of the image area and becomes disproportionately oversampled.

## 5 CONCLUSIONS

We presented 3DGEER, a complete, closed-form solution for projective-exact volumetric Gaussian rendering derived from first principles. To complement this formulation, we introduced an efficient PBF computation method for ray-particle association, enabling high-speed rendering while maintaining exactness in projective geometry. Additionally, we proposed the BEAP image representation to facilitate effective color supervision under wide-FoV camera models. Extensive experiments demonstrate that 3DGEER consistently outperforms state-of-the-art methods across multiple datasets, with particularly strong results in challenging wide-FoV scenarios.

ETHICS STATEMENT

This work builds on publicly available datasets (ScanNet++, ZipNeRF, Aria, and MipNeRF) and does not involve any human subjects, private data, or personally identifiable information. The methods developed are intended for advancing 3D scene reconstruction and rendering research, with potential applications in XR, robotics and autonomous driving. We do not foresee direct harmful consequences, but we note that any misuse of high-fidelity 3D reconstruction techniques could raise privacy concerns. To mitigate this, we restrict our experiments to research datasets that are widely used in the vision community.

REPRODUCIBILITY STATEMENT

We make significant efforts to ensure reproducibility. All mathematical derivations and proofs are provided in the Appendix (Secs. B, C and D). The datasets used in our experiments are publicly available (ScanNet++, ZipNeRF, Aria, and MipNeRF). Details of preprocessing, training protocols, and evaluation metrics are described in Sec. G. Our implementation will be released with training and evaluation scripts upon publication to further facilitate reproducibility.

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

CONTENTS

LLM Usage Statement

We used a large language model (ChatGPT, OpenAI) as a writing assistant. Specifically, the LLM was employed to (i) improve the clarity and readability of the manuscript, (ii) suggest stylistic edits to figure captions and section transitions, and (iii) provide alternative phrasings for technical descriptions without altering the scientific content.

The LLM was not involved in research ideation, algorithm design, data analysis, or experimental validation. All technical contributions, mathematical derivations, and results are the work of the authors. The authors take full responsibility for the content of this paper.

## A  Appendix: Exactness and Limitation

We clarify that our exactness refers to *eliminating linear approximation in projective geometry* during both the "rendering" and "ray-particle association" procedure, where errors will not decrease as the number of Gaussians increases under our settings (see Sec. 4.1). We do not claim exactness in the full physical volumetric sense. Our framework still involves approximations like ordering, self-attenuation, and overlapping which are our limitations.

**Limitations.** Our formulation integrates each Gaussian independently; hence it does not explicitly resolve self-occlusion, and ordering is handled through depth-based sorting. As also shown in the supplementary videos, this may lead to occasional popping artifacts. To our knowledge, EVER (Mai et al., 2025) is the only method that analytically resolves ordering and self-occlusion by approximating each Gaussian as a constant-density ellipsoid, though this comes with a substantial reduction in rendering speed.

Importantly, prior ray-tracing–based Gaussian renderers such as Fuzzy Metaballs (Keselman & Hebert, 2022) and subsequent methods including 3DGR/UT (Moënne-Loccoz et al., 2024; Wu et al., 2025) rely on the "maximum-response" assumption for ray–Gaussian interaction. This assumption inherently ignores ordering and self-attenuation, yet has been shown to introduce negligible error in practice (see Celarek et al. (2025) Fig. 9-11 / Sec. 7.1) regardless of the number of Gaussians. This explains that our depth-based ordering still can achieve the same practical accuracy as existing ray-tracing–based Gaussian renderers.

**Future work.**  While the practical impact of ordering and self-occlusion remains small in current Gaussian-based ray tracing, these factors fundamentally limit exact volumetric correctness. A promising future direction is to incorporate analytic or learnable self-attenuation models—potentially inspired by ellipsoidal integration (e.g., EVER Mai et al. (2025)) but without incurring heavy runtime costs. Another important avenue is handling ordering in a more principled manner, which may further reduce the rare popping artifacts observed under extreme configurations. We believe these limitations are orthogonal to our contributions on projective exactness, and addressing them would make the framework even more robust for large-scale or high-dynamic scenes.

## B  Appendix: Ray-Gaussian Transmittance with Projective Exactness

### B.1  Preliminaries And Notations

We follow the classic parameterization of 3D Gaussian primitives as introduced in prior work (Zwicker et al., 2001). Each 3D Gaussian distribution is defined by a mean position $\boldsymbol{\mu} \in \mathbb{R}^3$ and a 3D covariance matrix $\Sigma$ in the world coordinate system:

$$\mathcal{G}_{\Sigma,\boldsymbol{\mu}}(\mathbf{x}) = \frac{1}{\rho \, |\Sigma|^{1/2}} \exp\left( -\frac{1}{2} (\mathbf{x} - \boldsymbol{\mu})^\top \Sigma^{-1} (\mathbf{x} - \boldsymbol{\mu}) \right), \tag{B.1}$$

where we adopt the normalization constant $\rho$ as $\sqrt{2\pi}$, and $\mathbf{x} \in \mathbb{R}^3$ denotes a point in world space. This formulation differs from recent methods such as 3DGS (Kerbl et al., 2023) and 3DGUT (Wu et al., 2025), which omit the determinant term $|\Sigma|^{1/2}$ to simplify splatting or association during rendering. The form of our covariance matrix obeys: $\Sigma = RSS^\top R^\top$ , given a scaling matrix

$S = \mathrm{diag}(\mathbf{s})$ constructed from the scaling vector $\mathbf{s} \in \mathbb{R}^3$, and a rotation matrix $R \in \mathrm{SO}(3)$ derived from a unit quaternion $\mathbf{q} \in \mathbb{R}^4$.

To evaluate the transmittance contributed by a single 3D Gaussian particle along a ray $\mathbf{r}(t) = \mathbf{o} + t\mathbf{d}$, we have:

$$T(\mathbf{o}, \mathbf{d}) = \sigma \int_{t \in \mathbb{R}} \mathcal{G}_{\Sigma, \boldsymbol{\mu}}(\mathbf{o} + t\mathbf{d}) \, dt, \tag{B.2}$$

where $\sigma$ is the opacity coefficient associated with the Gaussian.

Finally, considering a set of 3D Gaussian particles sorted by depth contributing to the rendering color of the given ray, the rendered color $C$ is computed as:

$$C(\mathbf{o}, \mathbf{d}) = \sum_i c_i(\mathbf{o}, \boldsymbol{\mu}_i) T_i \prod_{j=1}^{i-1}(1 - T_j), \tag{B.3}$$

where $c_i(\mathbf{o}, \boldsymbol{\mu}_i)$ derives the view-dependent color of the $i$-th Gaussian as seen from the optical center $\mathbf{o}$, parameterized using spherical harmonics (SH) (Kerbl et al., 2023). Through minimizing the reconstruction loss between the rendered color and the observed color along each ray, the parameters $\mathbf{s}, \mathbf{q}, \boldsymbol{\mu}$, the coefficient $\sigma$ as well as the SH coefficients, are jointly optimized via backpropagation.

## B.2 Differentiable Rendering with Projective Exactness

The core challenge in achieving projective-exact Gaussian rendering in a ray marching formulation lies in obtaining a closed-form solution for integrating Gaussian density and color along a 3D ray. To address this, we first transform the ray—relative to each Gaussian—into a canonical coordinate system where the transmittance integral (see Eq. B.2) remains consistent and the Gaussians become isotropic. We then derive the exact ray color from first principles.

Specifically, we define a canonical coordinate system shared across all 3D Gaussian primitives. A point $\mathbf{u} \in \mathbb{R}^3$ in this canonical space is mapped to world coordinates for each Gaussian via:

$$\begin{bmatrix} \mathbf{x} \\ 1 \end{bmatrix} = H \begin{bmatrix} \mathbf{u} \\ 1 \end{bmatrix} = \begin{bmatrix} RS & \boldsymbol{\mu} \\ \mathbf{0} & 1 \end{bmatrix} \begin{bmatrix} \mathbf{u} \\ 1 \end{bmatrix}, \tag{B.4}$$

From this transformation, an infinitesimal segment $\Delta\mathbf{u}$ along a ray in the canonical frame corresponds to a segment $\Delta\mathbf{x} = RS\Delta\mathbf{u}$ in the world frame.

To ensure that the transmittance contributed by each Gaussian along the ray is preserved across coordinate systems, we require that the elementary transmittance (i.e., the product of density and segment length) remains invariant under the transformation (see Fig. 2 *Right*):

$$\mathcal{G}_{\mathbf{I}, \mathbf{0}}(\mathbf{u}) \, |\Delta\mathbf{u}| = \mathcal{G}_{\Sigma, \boldsymbol{\mu}}(\mathbf{x}) \, |\Delta\mathbf{x}| . \tag{B.5}$$

This constraint implies that the Jacobian determinant is absorbed into the measure. Note that $\left| \frac{\Delta\mathbf{x}}{\Delta\mathbf{u}} \right| = |RS| = |\Sigma|^{1/2}$. We can thus express the corresponding canonical density as:

$$\mathcal{G}_{\mathbf{I}, \mathbf{0}}(\mathbf{u}) = \frac{1}{\rho} \exp\left( -\frac{1}{2} \mathbf{u}^\top \mathbf{u} \right), \tag{B.6}$$

which is isotropic and shared across all primitives in canonical space.

This leads to an important conclusion that the exact closed-form solution for the transmittance integral of each Gaussian along the ray $\mathbf{r}(t)$ can thus be obtained by computing the accumulated density along the transformed ray $\mathbf{r}_u(t) = \mathbf{o}_u + t\mathbf{d}_u$ in the canonical isotropic Gaussian. Note that $t < 0$ is always negligible unless the optical center is within 3-sigma, then all 3D Gaussian rendering frameworks cull the Gaussian when optimization.

This admits a *closed-form* solution in terms of the perpendicular distance from the ray to the Gaussian center, as captured in Theorem 1:

$$T(\mathbf{o}, \mathbf{d}) = \sigma \int_{t \in \mathbb{R}} \mathcal{G}_{\mathbf{I}, \mathbf{0}}(\mathbf{o}_u + t\mathbf{d}_u) \, dt = \sigma \exp\left( -\frac{1}{2} \left( \mathrm{D}_{\mu, \Sigma}(\mathbf{o}, \mathbf{d}) \right)^2 \right), \tag{B.7}$$

where $D_{\mu,\Sigma}(\mathbf{o}, \mathbf{d})$ denotes the perpendicular distance from the Gaussian center (i.e., the origin in the canonical space) to the transformed ray $\mathbf{r}_u(t)$. Intuitively, this quantity also represents the minimal *Mahalanobis distance* from any points on the ray to the Gaussian in world space, measuring how close the ray passes by the particle distribution.

The remaining question is just about how to compute the perpendicular distance from the canonical origin to the transformed ray. We derive the transformed ray's direction vector $\mathbf{d}_u$ and moment vector $\mathbf{m}_u$ as:

$$\mathbf{d}_u = \left(S^{-1}R^\top\right)\mathbf{d}\,, \quad \mathbf{m}_u = \mathbf{o}_u \times \mathbf{d}_u, \tag{B.8}$$

where $\mathbf{o}_u = \left(S^{-1}R^\top\right)(\mathbf{o} - \boldsymbol{\mu})$ denotes the optical center in canonical space. Based on this, the square of $D_{\mu,\Sigma}(\mathbf{o}, \mathbf{d})$ can be efficiently calculated as:

$$\left(D_{\mu,\Sigma}(\mathbf{o}, \mathbf{d})\right)^2 = \frac{\mathbf{m}_u^\top \mathbf{m}_u}{\mathbf{d}_u^\top \mathbf{d}_u}. \tag{B.9}$$

By substituting this distance square into Eq. B.7, and then the derived transmittance value into Eq. B.3, we compute the accumulated color along the ray.

## B.3 RAY INTEGRAL OF ISOTROPIC 3D GAUSSIAN

**Theorem 1.** *Given a ray and a standard isotropic 3D Gaussian, the ray-Gaussian integral has a closed-form proportional to:*

$$\exp\left(-\frac{1}{2}D^2\right),$$

*where $D$ is the perpendicular distance from the Gaussian center to the ray.*

*Proof.* We first apply a translation to the coordinate system such that the Gaussian $\mathcal{G}$ is centered at the origin, and the ray $\mathbf{r}_u$ is translated accordingly. Next, we rotate the entire scene so that the ray direction is aligned with the $z$-axis. Due to the translation-invariance of the integral and the isotropy of the standard Gaussian, these transformations do not affect the result of the integral.

The translated standard isotropic 3D Gaussian is defined as:

$$\mathcal{G}_{\mathbf{I},\mathbf{0}}(\mathbf{u}) = \frac{1}{\rho}\exp\left(-\frac{1}{2}\mathbf{u}^\top \mathbf{u}\right), \tag{B.10}$$

which matches Eq. B.6. Let the $z$-axis aligned ray as:

$$\mathbf{r}_u(t) = \begin{bmatrix} \mathbf{o}_u \\ \mathbf{o}_v \\ 0 \end{bmatrix} + t\begin{bmatrix} 0 \\ 0 \\ 1 \end{bmatrix}, \tag{B.11}$$

the ray-Gaussian integral becomes:

$$\begin{aligned}
\int_{\mathbf{t}\in\mathbb{R}} \mathcal{G}_{\mathbf{I},\mathbf{0}}(\mathbf{r}_u(t))\, dt &= \frac{1}{\rho}\exp\left(-\frac{u_0^2 + v_0^2}{2}\right)\int_{t\in\mathbb{R}}\exp\left(-\frac{1}{2}t^2\right)\, dt \\
&= \frac{\sqrt{2\pi}}{\rho}\exp\left(-\frac{1}{2}D^2\right),
\end{aligned} \tag{B.12}$$

where $D^2 = \frac{u_0^2 + v_0^2}{2}$ denotes the squared perpendicular distance from the ray to the Gaussian center in canonical space. Thus, the integral is proportional to $\exp\left(-\frac{1}{2}D^2\right)$, as stated in theorem. $\square$

Note that the density function shared by our 3D primitives in the canonical space (see Eq. B.6) follows the same formulation as in the isotropic case. Given the world-coordinate ray $(\mathbf{o}, \mathbf{d})$ as the input of $D$, its value also depends on the world-space Gaussian parameters $\boldsymbol{\mu}, \Sigma$, which are used to transform the ray into the canonical coordinate system. Therefore, for each Gaussian contributing to the ray, we derive the transmittance as:

$$T = \frac{\sqrt{2\pi}\sigma}{\rho}\exp\left(-\frac{1}{2}\left(D_{\boldsymbol{\mu},\Sigma}(\mathbf{o}, \mathbf{d})\right)^2\right), \tag{B.13}$$

which recovers Eq. B.7 when picking $\rho = \sqrt{2\pi}$.

# C  APPENDIX: COOKBOOK OF GRADIENT COMPUTATION

## C.1  CHAIN RULE

Recall that $T = \sigma\alpha$ is the ray-Gaussian transmittance (Eq. B.7), where $\sigma$ is the opacity coefficient, and $\alpha$ is the ray-Gaussian integral. We denote $\kappa = D^2_{\mu,\Sigma}$ as the minimal squared Mahalanobis distance from the ray to the Gaussian distribution, $S$ as the Gaussian scaling, and $R$ as the rotation matrix in the world space. Additionally, given $\Sigma = RSS^\top R^\top$ as the 3D covariance matrix of the Gaussian in the world space, we have $W_{\mathrm{pca}}$ as the PCA whitening matrix satisfying:

$$W_{\mathrm{pca}} = \Sigma^{-\frac{1}{2}} = S^{-1}R^\top,$$
$$\text{where} \quad S^{-1} = \mathrm{diag}\left(\frac{1}{\mathbf{s}}\right). \tag{C.1}$$

As illustrated in Fig. 2, we can apply the chain rule to find derivatives *w.r.t.* the Gaussian parameters, e.g, scaling vector $\mathbf{s}$, rotation quaternion $\mathbf{q}$, and mean $\boldsymbol{\mu}$:

$$\frac{\partial T}{\partial \mathbf{s}} = \frac{\partial T}{\partial \kappa}\frac{\partial \kappa}{\partial W_{\mathrm{pca}}}\frac{\partial W_{\mathrm{pca}}}{\partial \mathbf{s}}, \tag{C.2}$$

$$\frac{\partial T}{\partial \mathbf{q}} = \frac{\partial T}{\partial \kappa}\frac{\partial \kappa}{\partial W_{\mathrm{pca}}}\frac{\partial W_{\mathrm{pca}}}{\partial \mathbf{q}}, \tag{C.3}$$

$$\frac{\partial T}{\partial \boldsymbol{\mu}} = \frac{\partial T}{\partial \kappa}\frac{\partial \kappa}{\partial \boldsymbol{\mu}}. \tag{C.4}$$

## C.2  TWO-STAGE PROPAGATION

To reduce computational overhead in shared propagation, we design our gradient computation following the structured two-stage approach of 3DGS (Kerbl et al., 2023), comprising `RenderCUDA` (RC) and `PreprocessCUDA` (PC). The former performs ray-wise accumulation, efficiently accelerated by our Particle Bounding Frustum (PBF) association (see Section 3.2), while the latter operates with Gaussian-wise complexity, leveraging view frustum culling for additional speedup.

According to Eq. B.7, we can respectively write the gradient *w.r.t.* opacity coefficient and the shared partial derivative for Gaussian parameters in `RC` as:

$$\left.\frac{\partial T}{\partial \sigma}\right|_{\mathrm{RC}} = \alpha, \quad \left.\frac{\partial T}{\partial \kappa}\right|_{\mathrm{RC}} = -\frac{T}{2}. \tag{C.5}$$

The remaining gradients to the Gaussian parameters can then be back-propagated from the squared distance, *e.g.*, $\frac{\partial \kappa}{\partial \mathbf{s}}, \frac{\partial \kappa}{\partial \mathbf{q}}, \frac{\partial \kappa}{\partial \boldsymbol{\mu}}$.

## C.3  GRADIENTS FOR CANONICAL RAY

We then backpropagate gradients to the canonical ray parameters: the origin $\mathbf{o}_u$, direction $\mathbf{d}_u$, and moment $\mathbf{m}_u = \mathbf{o}_u \times \mathbf{d}_u$, as defined in Eq. B.8 in our paper. These quantities directly influence the Mahalanobis distance in Eq. B.9 and are critical for capturing canonical ray geometry.

$$\left.\frac{\partial \kappa}{\partial \mathbf{m}_u}\right|_{\mathrm{RC}} = \left(\frac{2}{\mathbf{d}_u^\top \mathbf{d}_u}\right)\mathbf{m}_u, \tag{C.6}$$

$$\left.\frac{\partial \kappa}{\partial \mathbf{o}_u}\right|_{\mathrm{RC}} = \frac{\partial \kappa}{\partial \mathbf{m}_u} \times \mathbf{d}_u, \tag{C.7}$$

$$\left.\frac{\partial \kappa}{\partial \mathbf{d}_u}\right|_{\mathrm{RC}} = -\left(\frac{2\kappa}{\mathbf{d}_u^\top \mathbf{d}_u}\right)\mathbf{d}_u + \mathbf{o}_u \times \frac{\partial \kappa}{\partial \mathbf{m}_u}. \tag{C.8}$$

## C.4 GRADIENTS FOR WHITENING MATRIX

Recall that $\mathbf{d}_u = \mathrm{W}_{\mathrm{pca}}\mathbf{d}$ and $\mathbf{o}_u = \mathrm{W}_{\mathrm{pca}}(\mathbf{o} - \boldsymbol{\mu})$, where $\mathbf{o} = -R_c^\top \mathbf{t}_c$ denotes the world space optical center, and $\mathbf{d}$ denotes the ray direction. Once the gradients *w.r.t.* the ray parameters are obtained, we further propagate them to the PCA whitening matrix $\mathrm{W}_{\mathrm{pca}}$ via outer products with the world-space vectors:

$$
\begin{aligned}
\texttt{RenderCUDA:} \quad & \left.\frac{\partial \kappa}{\partial \mathrm{W}_{\mathrm{pca}}}\right|_{\mathrm{RC}} = \frac{\partial \kappa}{\partial \mathbf{d}_u}\mathbf{d}^\top ; \\
\texttt{PreprocessCUDA:} \quad & \left.\frac{\partial \kappa}{\partial \mathrm{W}_{\mathrm{pca}}}\right|_{\mathrm{PC}} = \frac{\partial \kappa}{\partial \mathbf{o}_u}(\mathbf{o} - \boldsymbol{\mu})^\top ; \\
\texttt{Overall:} \quad & \frac{\partial \kappa}{\partial \mathrm{W}_{\mathrm{pca}}} = \left.\frac{\partial \kappa}{\partial \mathrm{W}_{\mathrm{pca}}}\right|_{\mathrm{RC}} + \left.\frac{\partial \kappa}{\partial \mathrm{W}_{\mathrm{pca}}}\right|_{\mathrm{PC}} .
\end{aligned}
\tag{C.9}
$$

Here, the gradient backpropagated through $\mathbf{d}_u$ is accumulated ray-wise in $\texttt{RenderCUDA}$, while the gradient through $\mathbf{o}_u$ is computed Gaussian-wise in $\texttt{PreprocessCUDA}$. These two components are then summed to yield the complete gradient *w.r.t.* the PCA transformation matrix $\mathrm{W}_{\mathrm{pca}}$.

## C.5 GRADIENTS FOR SCALING AND ROTATION

Thus, for scaling gradients $\frac{\partial \kappa}{\partial \mathbf{s}} = \frac{\partial \kappa}{\partial \mathrm{W}_{\mathrm{pca}}}\frac{\partial \mathrm{W}_{\mathrm{pca}}}{\partial \mathbf{s}}$, we have:

$$
\left.\frac{\partial \mathrm{W}_{\mathrm{pca}}^{i,j}}{\partial s_k}\right|_{\mathrm{PC}} = \left\{ \begin{matrix} -\frac{1}{s_k^2}R_{j,k} & \text{if } i = k \\ 0 & \text{otherwise} \end{matrix} \right\} .
\tag{C.10}
$$

To derive gradients for rotation $\frac{\partial \kappa}{\partial \mathbf{q}} = \frac{\partial \kappa}{\partial \mathrm{W}_{\mathrm{pca}}}\frac{\partial \mathrm{W}_{\mathrm{pca}}}{\partial \mathbf{q}}$, given the quaternion form $\mathbf{q} = q_r + q_i\mathbf{i} + q_j\mathbf{j} + q_k\mathbf{k}$, we can write the whitening matrix as:

$$
\mathrm{W}_{\mathrm{pca}} = 2 \begin{pmatrix} \frac{1}{s_1}(\frac{1}{2} - q_j^2 - q_k^2) & \frac{1}{s_1}(q_iq_j + q_rq_k) & \frac{1}{s_1}(q_iq_k - q_rq_j) \\ \frac{1}{s_2}(q_iq_j - q_rq_k) & \frac{1}{s_2}(\frac{1}{2} - q_i^2 - q_k^2) & \frac{1}{s_2}(q_jq_k + q_rq_i) \\ \frac{1}{s_3}(q_iq_k + q_rq_j) & \frac{1}{s_3}(q_jq_k - q_rq_i) & \frac{1}{s_3}(\frac{1}{2} - q_i^2 - q_j^2) \end{pmatrix} .
\tag{C.11}
$$

As a result, we derive the following end gradients for the quaternion $\mathbf{q}$ as:

$$
\begin{aligned}
&\left.\frac{\partial \mathrm{W}_{\mathrm{pca}}}{\partial q_r}\right|_{\mathrm{PC}} = 2 \begin{pmatrix} 0 & \frac{q_k}{s_1} & -\frac{q_j}{s_1} \\ -\frac{q_k}{s_2} & 0 & \frac{q_i}{s_2} \\ \frac{q_j}{s_3} & -\frac{q_i}{s_3} & 0 \end{pmatrix}, \left.\frac{\partial \mathrm{W}_{\mathrm{pca}}}{\partial q_i}\right|_{\mathrm{PC}} = 2 \begin{pmatrix} 0 & \frac{q_j}{s_1} & \frac{q_k}{s_1} \\ \frac{q_j}{s_2} & -\frac{2q_i}{s_2} & \frac{q_r}{s_2} \\ \frac{q_k}{s_3} & -\frac{q_r}{s_3} & -\frac{2q_i}{s_3} \end{pmatrix}, \\
&\left.\frac{\partial \mathrm{W}_{\mathrm{pca}}}{\partial q_j}\right|_{\mathrm{PC}} = 2 \begin{pmatrix} -\frac{2q_j}{s_1} & \frac{q_i}{s_1} & -\frac{q_r}{s_1} \\ \frac{q_i}{s_2} & 0 & \frac{q_k}{s_2} \\ \frac{q_r}{s_3} & \frac{q_k}{s_3} & -\frac{2q_j}{s_3} \end{pmatrix}, \left.\frac{\partial \mathrm{W}_{\mathrm{pca}}}{\partial q_k}\right|_{\mathrm{PC}} = 2 \begin{pmatrix} -\frac{2q_k}{s_1} & \frac{q_r}{s_1} & \frac{q_i}{s_1} \\ -\frac{q_r}{s_2} & -\frac{2q_k}{s_2} & \frac{q_j}{s_2} \\ \frac{q_i}{s_3} & \frac{q_j}{s_3} & 0 \end{pmatrix} .
\end{aligned}
\tag{C.12}
$$

## C.6 GRADIENTS FOR 3D MEAN

Meanwhile, we compute end gradients *w.r.t.* the Gaussian 3D mean $\boldsymbol{\mu}$ as:

$$
\left.\frac{\partial \kappa}{\partial \boldsymbol{\mu}}\right|_{\mathrm{PC}} = -\mathrm{W}_{\mathrm{pca}}^\top \frac{\partial \kappa}{\partial \mathbf{o}_u} .
\tag{C.13}
$$

## C.7 NUMERICAL STABILITY OF OUR DIFFERENTIABLE FRAMEWORK

The choice of forward formulation, backward gradient computation, and intermediate cache variables critically affects numerical stability. Prior approaches such as GOF (Yu et al., 2024b) suffer from instability in backward propagation and therefore rely on additional 3D filtering heuristics to stabilize training. In contrast, our closed-form formulation, together with its exact gradient chain, avoids such instability altogether, eliminating the need for any auxiliary filtering.

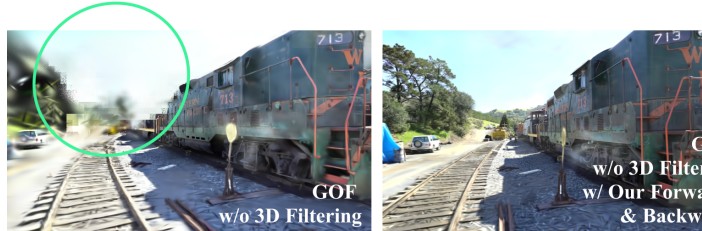

Figure C.1: **Artifacts without 3D filtering in GOF.** (*Left*) when degenerated Gaussians are not filtered in GOF, and (*Right*) the stable results obtained when applying our forward and backward formulations.

The key distinction from our gradient and rendering pipeline lies in that prior methods adopt an alternative expansion of $\mathbf{m}_u^\top \mathbf{m}_u$. Ideally, disregarding numerical issues, the following two forms are geometrically equivalent:

$$\mathbf{m}_u^\top \mathbf{m}_u = \|\mathbf{o}_u\|^2 \|\mathbf{d}_u\|^2 - (\mathbf{o}_u^\top \mathbf{d}_u)^2. \tag{C.14}$$

However, when a 3D Gaussian has one axis scaled close to zero, the canonical-space vectors $\mathbf{o}_u$ and $\mathbf{d}_u$ become nearly parallel. In this case, $\cos(\angle(\mathbf{o}_u, \mathbf{d}_u)) \to 1$, while both $\|\mathbf{o}_u\|$ and $\|\mathbf{d}_u\|$ diverge. This makes the term $\|\mathbf{o}_u\|^2 \|\mathbf{d}_u\|^2 - (\mathbf{o}_u^\top \mathbf{d}_u)^2$ highly prone to numerical overflow and, in practice, can even yield negative values and further lead to instable backward propagation as well.

Fig. C.1 (*Left*) demostrates the numerical artifacts produced by GOF. When it does not apply additional 3D filtering to remove degenerated Gaussians, severe artifacts appear. The snowflake noise arises from numerical overflow in Eq. C.14, where negative values are prematurely returned and extremely small positive values lead to amplified Gaussian intensity. Since the EWA approximation with Jacobian and conic remains numerically stable, the artifacts mainly stem from inaccurate maximum-response computation, which also induces gridline patterns.

In contrast, our CUDA rasterizer implementation computes both the cross product and its gradients directly (Eq. B.9 (forward) and Eqs. C.6–C.9 (backward)), thereby fundamentally avoiding this numerical issue. As a result, the forward pass is artifact-free, while the backward pass yields accurate gradients. Fig. C.1 (*Right*) illustrates the results of applying our forward and backward formulations to GOF: even without additional 3D filtering to exclude degenerated Gaussians (i.e., flat disk-like or needle Gaussians), our approach effectively eliminates the artifacts caused by numerical instability.

## D APPENDIX: ADDITIONAL DETAILS IN ASSOCIATION

Section D.1 provides our detailed derivation. Section D.2 provides implementation considerations required for stable PBF under large-FoV and omnidirectional cameras. Section D.3 clarifies the derivational relationships and explicitly states the conditions under which prior formulas coincide with ours.

### D.1 PARTICLE BOUNDING FRUSTUM (PBF) SOLVER

Recall Eq. 8, we define $P = (VH)^\top \in \mathbb{R}^{4 \times 4}$ as the plane transformation matrix from the view space to the canonical space. Take the unknown bounds for $c = \tan \theta_{1,2}$ as example, we have:

$$\mathbf{g}_u = -P_0 + cP_2 = \begin{bmatrix} -P_{00} + cP_{02} \\ -P_{10} + cP_{12} \\ -P_{20} + cP_{22} \\ -P_{30} + cP_{32} \end{bmatrix}, \tag{D.1}$$

where $P_i$ denotes $i$-th row of the matrix, expressed as a $4 \times 1$ column vector, and $P_{ij}$ further denotes $j$-th element. Recall Eq. 9, assuming we are computing the bounding box of the 1-standard-deviation contour ellipsoid of the Gaussian, *i.e.*, $\lambda = 1$, then we have the following quadratic equation:

$$\left( (\mathbf{f} \circ P_2^\top) \cdot P_2 \right) c^2 - 2 \left( (\mathbf{f} \circ P_0^\top) \cdot P_2 \right) c + (\mathbf{f} \circ P_0^\top) \cdot P_0 = 0, \tag{D.2}$$

where $\mathbf{f} = (1, 1, 1, -1)$ is a $1 \times 4$ row vector. Here we denote the equation as:

$$\mathcal{T}_{22}c^2 - 2\mathcal{T}_{02}c + \mathcal{T}_{00} = 0, \tag{D.3}$$

where $\mathcal{T}$ is a $3 \times 3$ upper triangular matrix defined with $\mathcal{T}_{ij} = \left(f \circ P_i^\top\right) \cdot P_j$.

Similarly, for the bounds of $\phi$, we have $c = \tan\phi_{1,2}$ satisfying:

$$\mathcal{T}_{22}c^2 - 2\mathcal{T}_{12}c + \mathcal{T}_{11} = 0. \tag{D.4}$$

Interestingly, given a camera-view configuration $[R_c \mid \mathbf{t}_c]$, we find that the parameter matrix $\mathcal{T}$ can be efficiently computed from the Gaussian's view-space covariance $\Sigma_c = R_c R S S^\top R^\top R_c^\top$ and 3D mean $\boldsymbol{\mu}_c = R_c \boldsymbol{\mu} + \mathbf{t}_c$:

$$\mathcal{T}_{ij} = \lambda^2 \Sigma_c^{i,j} - \left(\boldsymbol{\mu}_c \boldsymbol{\mu}_c^\top\right)^{i,j}. \tag{D.5}$$

Thus, we can compute the angular bounds (defined by their tangent-space center and extent) from:

$$\left(\frac{\tan\theta_1 + \tan\theta_2}{2}, \frac{\tan\phi_1 + \tan\phi_2}{2}\right) = \left(\frac{\mathcal{T}_{02}}{\mathcal{T}_{22}}, \frac{\mathcal{T}_{12}}{\mathcal{T}_{22}}\right),$$

$$\|\tan\theta_1 - \tan\theta_2\| = \frac{2\sqrt{\mathcal{T}_{02}^2 - \mathcal{T}_{22}\mathcal{T}_{00}}}{|\mathcal{T}_{22}|}, \tag{D.6}$$

$$\|\tan\phi_1 - \tan\phi_2\| = \frac{2\sqrt{\mathcal{T}_{12}^2 - \mathcal{T}_{22}\mathcal{T}_{11}}}{|\mathcal{T}_{22}|}.$$

If the discriminant of either quadratic is negative—indicating that the camera's optical center lies inside the Gaussian ellipsoid and the angular bounds are undefined—we conservatively clamp the frustum angles using the camera's maximum field of view (FoV).

Note that this result is directly computed from the 3D view-space true covariance, in contrast to methods like EWA and UT, which rely on intermediate conic approximations in screen space. Our PBF formulation offers both exactness and efficiency.

## D.2 DETAILS IN COMPUTING CSF-PBF INTERSECTION

We uniformly sample spherical angles within the maximum field of view (FoV). The resulting $x$, $y$, and $z$ coordinates—i.e., the ray direction vectors $\mathbf{d}$—are projected onto a unit sphere and then transformed into pixel space for inverse color interpolation (see Sec. E). These sampled rays are then grouped according to their corresponding CSFs for subsequent Gaussian association.

In most scenes using a pinhole camera with a limited FoV, the angular bounds in PBF typically lie within $[-\frac{\pi}{2}, \frac{\pi}{2}]$. In such cases, since the $\tan$ function is monotonically increasing over this interval, the intersections between PBF and CSFs can be computed by directly comparing the $\tan$ values of their respective angular bounds. However, in wide FoV scenarios, even when the incidence angle formed by the Gaussian centers remains within $[-\frac{\pi}{2}, \frac{\pi}{2}]$, the angular bounds used in PBF may still fall outside the monotonic region of the $\tan$ function. In these cases, directly comparing $\tan$ values may yield incorrect PBF-CSF intersections, such as mistakenly matching angular bounds as $[\theta_2', \theta_1]$ (see Fig. D.1 for an example).

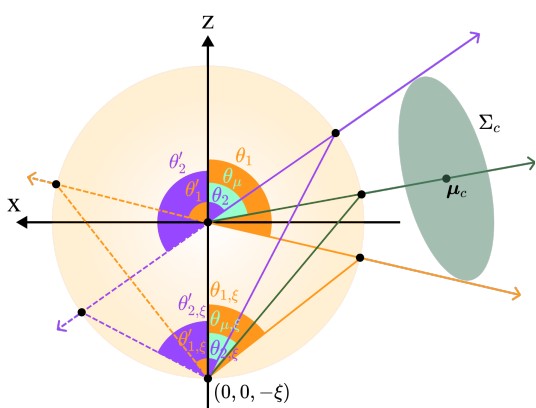

Figure D.1: **Mirror Transformation of PBF Angular Bounds.** Four candidate mirrored angles $\theta_{1,\xi}, \theta_{1,\xi}', \theta_{2,\xi}, \theta_{2,\xi}'$ are generated from a given pair of bounds. Given a Gaussian centered at $\boldsymbol{\mu}_c$, the two closest to the corresponding Gaussian's mirrored angle $\theta_{\mu,\xi}$ are used as PBF bounds for robust intersection with mirror-transformed CSF partitions.

To resolve the correct intersections, we introduce a mirror transformation applied to the tangent values of both PBF angular bounds and CSF partitions. Inspired by the CMEI camera model (Mei & Rives, 2007), we shift the optical center using a mirror parameter $\xi$, leading to the following transformation:

$$\tan\theta_\xi = \frac{\tan\theta}{1 + \xi\frac{z}{\|z\|}\sqrt{1+\tan^2\theta}} = \frac{\sin\theta}{\cos\theta + \xi}, \tag{D.7}$$

where $\theta_\xi$ represents the mirror-transformed angle of $\theta$ on the $xz$-plane, using a mirror parameter $\xi$. We set $\xi = 1$ throughout all experiments.

Given a pair of tangent value $[\tan\theta_1, \tan\theta_2]$ from our PBF solver (See Equation. D.6), this transformation yields four candidate mirror angles, as illustrated in Fig. D.1. The Gaussian center, in contrast, corresponds to a unique mirror-transformed angle $\theta_{\mu,\xi}$. We exploit this property by selecting the two candidate angles closest to $\theta_{\mu,\xi}$. These two then define the effective lower and upper bounds for PBF, i.e., $[\theta_{1,\xi}, \theta_{2,\xi}]$, which are subsequently compared with CSF partitions to compute the correct intersections.

This trick ensures robust and consistent angular comparisons, even beyond the monotonic domain of the `tan` function. As reported in our timing analysis (Tab. K.5), the pre-processing pipeline—including the mirror-angle-based PBF-CSF intersection computation and the canonical transformation for all Gaussians—takes less than 0.13 ms in total. This confirms the efficiency of our approach, even when applied to thousands of Gaussians across wide FoV scenarios.

### D.3 RELATION TO PRIOR WORKS

Substituting the perspective screen space coordinates:

$$u = \tan\theta, \qquad v = \tan\phi$$

into our formulation yields expressions identical to those in 2DGS (Huang et al., 2024a) and HTGS (Hahlbohm et al., 2025). This is expected as a perspective alternative of our angular-domain formulation.

Concretely, 2DGS and HTGS start from a screen-space or tangent-plane parameterization and therefore rely on assumptions that hold only for perspective (pinhole) cameras, e.g., geometric identities that arise when rays are implicitly tied to a planar, z-parallel image plane. This assumption holds only for perspective cameras and does not extend to fisheye or omnidirectional systems, where the corresponding surface in screen space is curved. In contrast, our derivation begins from a camera-agnostic angular parameterization in 3D and directly parameterizes Camera Sub-Frustums (CSFs) and Particle Bounding Frustums (PBFs). These different starting assumptions change the known quantities and constraints available during derivation, and thus the mathematical conditions under which closed-form bounds and culling guarantees remain valid for generic cameras.

We'd like to emphasize that mathematics in itself is a general tool, but deriving these bounds under weaker assumptions (i.e., without relying on screen-space structure) and validating them with comprehensive experiments across both pinhole and wide-FOV camera families constitute an essential part of our contribution.

This difference in parameterization also affects the downstream computational pipeline. Gaussian association, tiling/duplication, and sorting depend on how Gaussians map to tiles or bins. Using an angular (CSF-based) domain yields a different bucketing structure compared to pinhole screen-space tiling. This is practically important for large-FoV scenes: under generic cameras a Gaussian's bound may appear partially "behind" the camera in screen coordinates (see Fig. D.1), causing screen-space methods to produce mirrored or incorrect tangent-based bounds (dashed line in Fig. D.1). To robustly handle these cases, we introduce mirror parameters in the association step (Section D.2) and incorporate the Gaussian center into sorting and correction—components unnecessary in perspective-only derivations.

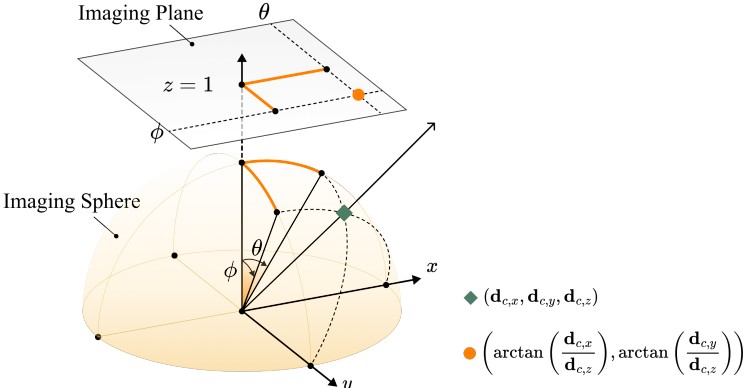

Figure E.1: **Spherical Angles Defining PBF and BEAP**. A view-space direction $\mathbf{d}_c$ is first normalized onto the optical sphere (green spade), followed by angular projections onto the $xz$ and $yz$ planes. The orange point denotes the resulting coordinate $(\theta, \phi)$ on the unfolded BEAP imaging plane.

# E   APPENDIX: BEAP IMAGE INVERSE INTERPOLATION

To prepare training data, we sample rays to comprehensively cover the original image FoVs and use them as input to our CUDA renderer. Simultaneously, we interpolate the ground truth color for each input ray to ensure accurate and uniform supervision during training.

The projection of sampled rays onto observed images begins with transforming the incidence angle projections $\theta$ and $\phi$ into normalized ray direction vectors $\mathbf{d} = (x, y, z)^\top$, expressed as:

$$x = \frac{\sin\theta\cos\phi}{\sqrt{\sin^2\theta\cos^2\phi + \cos^2\theta\sin^2\phi + \cos^2\theta\cos^2\phi}}, \tag{E.1}$$

$$y = \frac{\cos\theta\sin\phi}{\sqrt{\sin^2\theta\cos^2\phi + \cos^2\theta\sin^2\phi + \cos^2\theta\cos^2\phi}}, \tag{E.2}$$

$$z = \frac{\cos\theta\cos\phi}{\sqrt{\sin^2\theta\cos^2\phi + \cos^2\theta\sin^2\phi + \cos^2\theta\cos^2\phi}}. \tag{E.3}$$

Several camera models are provided to subsequently project the normalized rays $(x, y, z)^\top$ into pixel space $(x_p, y_p, 1)^\top$.

**Pinhole Camera Model.**   For pinhole camera models, we first have $\frac{x}{z} = \tan\theta$ and $\frac{y}{z} = \tan\phi$. Thus, we have:

$$\begin{bmatrix} x_p \\ y_p \\ 1 \end{bmatrix} = \begin{bmatrix} f_x & 0 & c_x \\ 0 & f_y & c_y \\ 0 & 0 & 1 \end{bmatrix} \begin{bmatrix} \tan\theta \\ \tan\phi \\ 1 \end{bmatrix}. \tag{E.4}$$

**KB Fisheye Model.**   We have the incidence angle $\varphi$ derived from its projections:

$$r = \sqrt{\tan^2\theta + \tan^2\phi} \quad \text{and} \quad \varphi = \arctan(r). \tag{E.5}$$

And we have the fisheye distorted angle:

$$\varphi_d = \varphi(1 + k_1\varphi^2 + k_2\varphi^4 + k_3\varphi^6 + k_4\varphi^8). \tag{E.6}$$

As a result, the rays are projected onto pixel space, allowing for inverse color interpolation from:

$$\begin{bmatrix} x_p \\ y_p \\ 1 \end{bmatrix} = \begin{bmatrix} \frac{f_x\varphi_d}{r} & 0 & c_x \\ 0 & \frac{f_y\varphi_d}{r} & c_y \\ 0 & 0 & 1 \end{bmatrix} \begin{bmatrix} \tan\theta \\ \tan\phi \\ 1 \end{bmatrix}. \tag{E.7}$$

## F  APPENDIX: HIGH-LEVEL ALGORITHMIC SUMMARY

We provide a high-level overview of our volumetric rendering pipeline (See Alg. 1 and Alg. 2). For clarity, we follow the structure of 3DGS Kerbl et al. (2023) to illustrate differences: newly introduced modules or replaced ones are highlighted in **green**, and significantly modified ones in **yellow**.

---

**ALGORITHM 1:** GPU Software Asso. & Render.

**Function** 3DGEER ($\boldsymbol{\mu}, \mathbf{s}, \mathbf{q}, c, \sigma, \mathbf{r}_c, V$):

$\quad$ CullGaussian($\boldsymbol{\mu}, V$) $\qquad\qquad\qquad\qquad$ ▷ Frustum Culling

$\quad$ $F \leftarrow$ InitCSFs($w, h, K$) $\qquad\qquad\qquad$ ▷ **Partition**

$\quad$ $\Sigma_c, \boldsymbol{\mu}_c \leftarrow$ ViewspaceGaussians($\mathbf{s}, \mathbf{q}, V$) $\qquad$ ▷ **Transform**

$\quad$ $F_{ID} \leftarrow$ PBF_Intersect($\Sigma_c, \boldsymbol{\mu}, F$) $\qquad\qquad$ ▷ **Sub-frustum Asso.**

$\quad$ $\mathrm{W}_{\mathrm{pca}} \leftarrow$ PCA_Whitening($\mathbf{s}, \mathbf{q}$) $\qquad\qquad$ ▷ **Transform**

$\quad$ $\mathcal{G}_{ID}, F_K \leftarrow$ DuplicateWithKeys($\mathrm{W}_{\mathrm{pca}}, F_{ID}$)

$\qquad\qquad\qquad\qquad$ ▷ **Indices (Gaussian) & Keys (FrustumID+Depth)**

$\quad$ SortByKeys($\mathcal{G}_{ID}, F_K$) $\qquad\qquad\qquad$ ▷ Globally Depth Sort

$\quad$ $R \leftarrow$ IdentifyFrustumRanges($F, F_K$)

$\quad$ $I \leftarrow \mathbf{0}$ $\qquad\qquad\qquad\qquad\qquad$ ▷ **Init Canvas (w/ Rays)**

$\quad$ **for all** CSFs $f \leftarrow I$ **do**

$\quad\quad$ **for all** Rays $i \leftarrow f$ **do**

$\quad\quad\quad$ $r \leftarrow$ GetFrustumRange($R, f$)

$\quad\quad\quad$ $\mathbf{m}_u, \mathbf{d}_u \leftarrow$ Cano_Ray($\mathrm{W}_{\mathrm{pca}}, \boldsymbol{\mu}, i$) $\qquad$ ▷ **Transform**

$\quad\quad\quad$ $\mathrm{D}_{\boldsymbol{\mu}, \Sigma} \leftarrow$ Dist($\mathbf{m}_u, \mathbf{d}_u$) $\qquad\qquad$ ▷ **Mah-distance**

$\quad\quad\quad$ $I[i] \leftarrow$ BlendInOrder($i, \mathcal{G}_{ID}, r, F_K, \mathrm{D}_{\boldsymbol{\mu}, \Sigma}, c, \sigma$)

$\qquad\qquad\qquad\qquad\qquad\qquad$ ▷ Alpha Blending

$\quad\quad$ **end**

$\quad$ **end**

$\quad$ **return** $I$

---

**ALGORITHM 2:** 3DGEER Initialization and Training Framework

$w, h$: width and height of the training images

$K, V$: intrinsics and extrinsic meta info.

$\qquad\qquad\qquad\qquad\qquad\qquad\qquad\qquad$ ▷ Init Gaussian Attributes

$\boldsymbol{\mu} \leftarrow$ SfM Points $\qquad\qquad\qquad\qquad\qquad\qquad$ ▷ Position

$\mathbf{s}, \mathbf{q} \leftarrow$ InitAttributes() $\qquad\qquad\qquad\qquad$ ▷ Scaling, Rotation

$c, \sigma \leftarrow$ InitCoefficients() $\qquad\qquad\qquad\qquad$ ▷ Color, Opacity

**while** not converged **do**

$\quad$ $\mathbf{r}_c, \hat{C} \leftarrow$ BEAP($w, h, K$) $\qquad\qquad\qquad$ ▷ **View Rays / Colors**

$\qquad\qquad\qquad\qquad\qquad\qquad\qquad\qquad$ ▷ **Asso. & Render**

$\quad$ $C \leftarrow$ 3DGEER($\boldsymbol{\mu}, \mathbf{s}, \mathbf{q}, c, \sigma, \mathbf{r}_c, V$) $\qquad\qquad$ ▷ See Alg. 1

$\quad$ $\mathcal{L} \leftarrow$ Loss($C, \hat{C}$) $\qquad\qquad\qquad\qquad\qquad$ ▷ Loss

$\quad$ $\boldsymbol{\mu}, \mathbf{s}, \mathbf{q}, c, \sigma \leftarrow$ Adam($\nabla\mathcal{L}$) $\qquad\qquad$ ▷ **Backprop & Step**

$\quad$ **if** IsRefinementIteration($i$) **then**

$\quad\quad$ $\boldsymbol{\mu}, \mathbf{s}, \mathbf{q}, c, \sigma \leftarrow$ Optimize($\boldsymbol{\mu}, \mathbf{s}, \mathbf{q}, c, \sigma$) $\qquad$ ▷ Kerbl et al. (2023)

$\quad$ **end**

**end**

---

## G  APPENDIX: ADDITIONAL IMPLEMENTATION DETAILS

**Baselines.** We compare 3DGEER with leading volumetric particle rendering methods targeting projective approximation errors in 3DGS (Kerbl et al., 2023). For large FoV fisheye inputs, we evaluate against FisheyeGS (Liao et al., 2024), EVER (Mai et al., 2025), and 3DGUT (Wu et al., 2025). Under pinhole settings, we additionally compare with 3DGS and 3DGRT (Moënne-Loccoz et al., 2024).

**Implementation Details.** To ensure fair comparison, we align our implementation with the original 3DGS in terms of initialization and hyperparameters on the ScanNet++ and MipNeRF360 datasets. For ZipNeRF, we adopt the training configurations of EVER and SMERF (Duckworth et al., 2024) for all candidate methods. Similar to most ray-based frameworks, our method does not directly access gradients of `mean2D` to control densification or pruning. Instead, we optimize using gradients of the view-space `mean3D`. Unless otherwise noted, all models are trained with consistent supervision strategies and architectural settings. For fisheye datasets (ScanNet++ and ZipNeRF), we precompute bijective grid mappings (Guo et al., 2025) and reproject all rendered outputs to the native KB-Fisheye image space (Kannala & Brandt, 2006) before evaluation. Experiments on ScanNet++ and MipNeRF360 are conducted using full-resolution images. For ZipNeRF, due to the high frame count and extremely large fisheye resolution, we use the maximum resolution (see Tab. 3) that allows all baselines to run reliably on a local desktop setup.

**Central and peripheral regions in ScanNet++ evaluation.** The rays sampled in the central part follows the range:

$$\theta \in [-\frac{\text{w}}{2f_x}, \frac{\text{w}}{2f_x}], \phi \in [-\frac{\text{h}}{2f_y}, \frac{\text{h}}{2f_y}].$$

And the Peripheral part takes the rays sampled in remaining angle ranges. Here $\text{w}, \text{h}$ indicate the width and height of the image and $f_x, f_y$ are the focal lengths defined in the original KB camera model.

**Cross-camera generalization evaluation on ZipNeRF.** ZipNeRF provides two sets of data: fisheye raw images and their corresponding undistorted frames, which are generated by applying distortion correction and cropping the central regions. Although the frames are aligned on a per-frame basis, camera poses and point clouds were independently reconstructed using COLMAP (Schönberger & Frahm, 2016) for each set. Due to COLMAP's global optimization independently refining camera trajectories, applying a single rigid transformation to align the two pose sets is insufficient and may introduce alignment errors. To ensure reliable cross-camera evaluation without introducing errors from structure-from-motion (SfM), we always use the same set of test-time camera poses as those used during training. In cross-camera experiments, we vary only the intrinsic parameters, while keeping the extrinsic poses fixed. This ensures fair comparisons without entangling pose inconsistencies arising from preprocessing.

## H APPENDIX: EWA SPLATTING PRELIMINARIES

**From camera coordinates to ray coordinates.** As the coordinate definitions in EWA Splatting (Zwicker et al., 2001), we have the following mapping function from the camera (view) space points $\mathbf{t}$ to the ray space points $\mathbf{x}$ for projective transformation in pinhole camera models, *i.e.*, $\tau : \mathbf{t} \xrightarrow{\sim} \mathbf{x}$:

$$\mathbf{x} = \begin{bmatrix} x_0 \\ x_1 \\ x_2 \end{bmatrix} = \tau(\mathbf{t}) = \begin{bmatrix} t_0/t_2 \\ t_1/t_2 \\ \left\| (t_0, t_1, t_2)^\top \right\| \end{bmatrix}, \tag{H.1}$$

$$\mathbf{t} = \begin{bmatrix} t_0 \\ t_1 \\ t_2 \end{bmatrix} = \tau^{-1}(\mathbf{x}) = \begin{bmatrix} x_0/l \cdot x_2 \\ x_1/l \cdot x_2 \\ 1/l \cdot x_2 \end{bmatrix}, \tag{H.2}$$

$$, \text{ where } l = \left\| (x_0, x_1, 1)^\top \right\|. \tag{H.3}$$

These mappings are not affine, which requires computationally expensive (Westover, 1990; Mao et al., 1995) or locally approximated solutions (Zwicker et al., 2001; Kerbl et al., 2023) to calculate integral in the ray space.

**Local affine approximation.** EWA splatting by Zwicker et al. (2001) introduced local affine approximation, denoted as $\tau_k$, to keep the linear effects and the $k$-indexed reconstruction kernels as Gaussians in 2D screen space. It is defined by the first two terms of the Taylor expansion of $\tau$ at the camera space point $\mathbf{t}_k$, then the ray space point $\mathbf{x}$ near around $\mathbf{x}_k$ can be estimated by:

$$\mathbf{x} = \tau(\mathbf{t}) = \mathbf{x}_k + \mathbf{J}_\tau^{\mathbf{t}_k}(\mathbf{t} - \mathbf{t}_k), \tag{H.4}$$

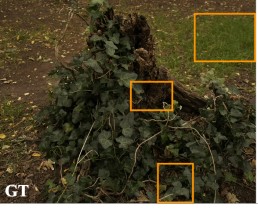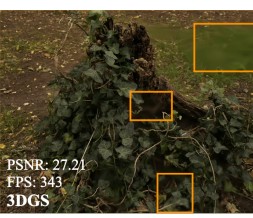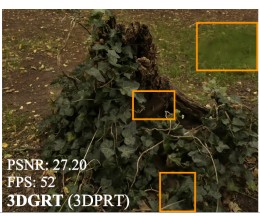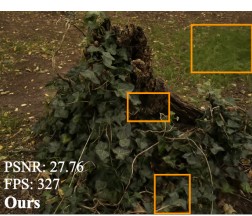

Figure I.1: **Qualitative Results on the Stump (Mip-NeRF360).** 3DGEER outperforms both 3DGS-based (Gaussian Splatting) and 3DPRT-based (Particle Ray Tracing) methods on novel view synthesis benchmarks under (*Left*) MipNeRF360 (pinhole), while maintaining high runtime efficiency. Larger differences are visible in the colored boxes.

where $\mathbf{J}_\tau^{\mathbf{t}_k}$ is the Jacobian matrix given by the partial derivatives of $\tau$ at the point $\mathbf{t}_k$. For the $k$-indexed 3D Gaussian $\mathcal{G}_{\Sigma,\boldsymbol{\mu}}$ in a certain camera coordinate, using EWA local affine transformation, the approximated density function can be expressed as a 3D Gaussian $\mathcal{G}_{\Sigma_k,\mathbf{x}_k}$ in the ray coordinate. For $\forall \mathbf{x}$ around $\mathbf{x}_k$, we have:

$$\mathcal{G}_{\Sigma_k,\mathbf{x}_k}(\mathbf{x}) = \frac{1}{(2\pi)^{\frac{3}{2}}|\Sigma_k|^{\frac{1}{2}}} \exp\left(-\frac{1}{2}(\mathbf{x} - \mathbf{x}_k)^\top \Sigma_k^{-1}(\mathbf{x} - \mathbf{x}_k)\right), \quad (\text{H.5})$$

$$\text{where } \Sigma_k = \mathbf{J}_\tau^{\mathbf{t}_k} \Sigma \left(\mathbf{J}_\tau^{\mathbf{t}_k}\right)^\top. \quad (\text{H.6})$$

Integrating a 3D Gaussian $\mathcal{G}_{\Sigma_k,\mathbf{x}_k}^{(3)}$ along one coordinate axis yields a 2D Gaussian $\mathcal{G}_{\hat{\Sigma}_k,\hat{\mathbf{x}}_k}^{(2)}$, hence:

$$\int_{\mathbb{R}} \mathcal{G}_{\Sigma_k,\mathbf{x}_k}^{(3)}(\mathbf{x})\, dx_2 = \mathcal{G}_{\hat{\Sigma}_k,\hat{\mathbf{x}}_k}^{(2)}(\hat{\mathbf{x}}), \quad (\text{H.7})$$

where $\hat{\mathbf{x}} = \begin{bmatrix} x_0 \\ x_1 \end{bmatrix}$, and the mean $\hat{\mathbf{x}}_k = \begin{bmatrix} x_{k,0} \\ x_{k,1} \end{bmatrix}$, and the $2 \times 2$ variance matrix $\hat{\Sigma}_k$ is easily obtained from the $3 \times 3$ matrix $\Sigma_k$ by skipping the third row and column.

**Splatting approximation error.** Although we can not expect the density function in ray space still obey the Gaussian distribution when we apply general projective transformation to the original view space, exact density can be derived through substitution via Eq. H.2:

$$\mathcal{G}_{\Sigma,\boldsymbol{\mu}}(\mathbf{t}) = \mathcal{G}_{\Sigma,\boldsymbol{\mu}}\left(\tau^{-1}(\mathbf{x})\right)$$
$$= \frac{1}{(2\pi)^{\frac{3}{2}}|\Sigma_k|^{\frac{1}{2}}} \exp\left(-\frac{1}{2}\left(\tau^{-1}(\mathbf{x}) - \mathbf{t}_k\right)^\top \Sigma_k^{-1}\left(\tau^{-1}(\mathbf{x}) - \mathbf{t}_k\right)\right), \quad (\text{H.8})$$

which means that the approximation error can be estimated through Monte Carlo integration of the following equation:

$$\mathcal{E}_k = \int_{\mathbb{R}} \left|\mathcal{G}_{\Sigma,\boldsymbol{\mu}}\left(\tau^{-1}(\mathbf{x})\right) - \mathcal{G}_{\Sigma_k,\mathbf{x}_k}(\mathbf{x})\right|\, dx_2. \quad (\text{H.9})$$

Note that $x_2$ represents the dimension along the ray direction in the ray coordinate.

# I  APPENDIX: ADVANTAGE OVER PRIOR RAY-TRACING METHODS

Pure ray-tracing (e.g., 3DGRT) can be ideally projective-exact, but in practice the BVH adopted in 3DGRT for intersection tests uses inscribed proxy icosahedron for faster speed, which is tight at vertices but face centers lie ∼20% inward based on icosahedron's property. Proxy shapes may degrade intersection accuracy and further give worse accumulated gradients during density control and eventually lead to incorrect geometry (e.g., missing leaves in Fig. I.1). Further, intersection tests for far, tiny, and cluttered Gaussians are limited by floating-point precision that causes noisy rendering. (E.g., in Fig. 15 of the 3DGRT paper, native rendering in the leftmost block contains visible speckles in the tree and background areas.) Even 3DGUT with UT approximation outperforms 3DGRT on

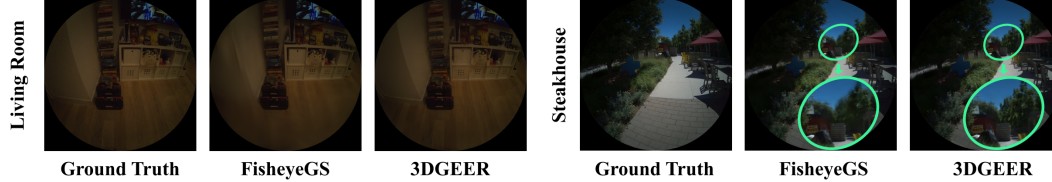

Figure J.1: **Qualitative Results on the Aria Dataset.** Our method produces sharper floor textures in indoor scenes and clearer distant tree details in outdoor scenes.

Table J.1: **Quantitative comparison on Aria**. Higher PSNR/SSIM and lower LPIPS are better.

| Method | Scene | | PSNR↑ | SSIM↑ | LPIPS↓ |
|---|---|---|---|---|---|
| 3DGEER (Ours) | livingroom
steakhouse | (indoor)
(outdoor) | **36.398**
**30.662** | **0.933**
**0.881** | **0.226**
**0.246** |
| FisheyeGS | livingroom
steakhouse | (indoor)
(outdoor) | 35.686
29.989 | 0.927
0.869 | 0.231
0.261 |

MipNeRF360. For validation, we use 3DGRT's ray-tracing and rendering engine and our 3DGEER rendering pipeline to render from the same pretrained MipNeRF360-stump and compare on PSNR↑/ SSIM↑/ LPIPS↓: 3DGRT- 25.79/ 0.745/ 0.264; ours- 26.83/ 0.774/ 0.246 (+1.04/ +0.029/ -0.018). 3DGEER builds a projective-exact and efficient pipeline and further and optimizes with BEAP color supervision, achieving much faster speed and better accuracy (Tab. 4).

## J  APPENDIX: RESULTS ON EGOCENTRIC DATASET

Recent egocentric devices typically feature wide FoVs, but existing pipelines struggle to maintain reconstruction quality when using full-resolution images or when restricted to cropped views. The challenge is further compounded by low-light imaging conditions—where limited light intake, especially with small sensors and aperture-constrained lenses, exacerbates degradation.

We evaluate on the Aria egocentric dataset ($\sim 110°$ FoV, circular shape). Our method achieves an average PSNR of 33.5dB, consistently surpassing FisheyeGS (32.8dB) across all metrics (Tab. J.1). Qualitatively, our reconstructions preserve sharper details in peripheral regions for the indoor scene (Fig. J.1, *Left*) and recover distant structures such as trees in the outdoor scene (Fig. J.1, *Right*). The performance gain (+0.7 dB) lies between the improvements observed in pinhole datasets (e.g., MipNeRF360: +0.3dB; ZipNeRF-cross: +2.0dB) and in highly distorted wide-FoV datasets (e.g., ScanNet++/ZipNeRF $180°$ diagonal FoV: +0.9–4.8dB), which aligns with Aria's moderate FoV and distortion level.

# K APPENDIX: ADDITIONAL MATERIAL AND RESULTS

## K.1 ADDITIONAL MATERIAL.

Code, video, and an HTML page are provided in the Supplementary Material.

## K.2 ADDITIONAL RESULTS.

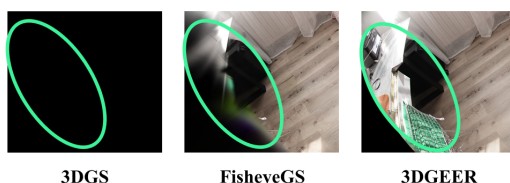

Figure K.1: **Out-of-Distribution (OOD) Region in Fig. 6.** 3DGS collapses to black due to the numerical issue of $\tan 90° \to \infty$, while our method robustly renders views up to $180°$, sucessfully to reconstruct the green chair, where splat-based methods (e.g., FisheyeGS) fail. Similarly, in Fig. 6 (zoom-in), our method also preserves structures like the corridor and fridge, table and door, or window and cabinet.

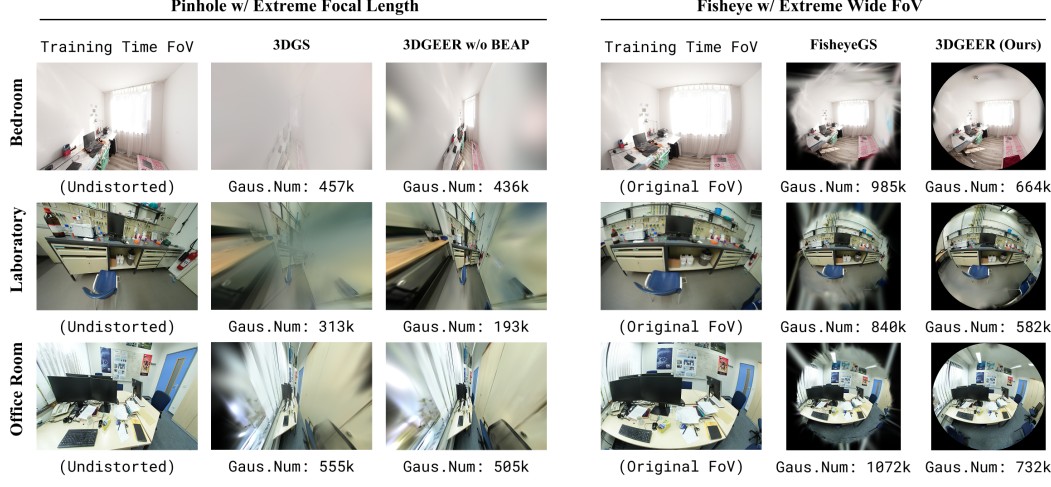

Figure K.2: **Extreme-FoV Generalization.** (*Left*) Comparison between our method and 3DGS under extreme-FoV pinhole $(170°)$ views (trained on undistorted central regions). (*Right*) Zoom-out results of Fig. 6, showing the complete circular $180°$ FoV, where our method generalizes robustly to out-of-distribution regions.

Table K.1: **MipNeRF360 Quantitative Results by Sequence.** The overall results are shown in Tab. 4.

| Method | Metric | Bicycle | Bonsai | Counter | Garden | Kitchen | Stump | Flowers | Room | Treehill |
|--------|--------|---------|--------|---------|--------|---------|-------|---------|------|----------|
| Ours | PSNR↑ | 25.63 | 32.23 | 29.05 | 27.60 | 31.78 | 26.83 | 21.83 | 32.09 | 22.83 |
| | SSIM↑ | 0.774 | 0.946 | 0.910 | 0.860 | 0.931 | 0.774 | 0.613 | 0.946 | 0.637 |
| | LPIPS↓ | 0.223 | 0.175 | 0.195 | 0.136 | 0.120 | 0.246 | 0.340 | 0.108 | 0.344 |

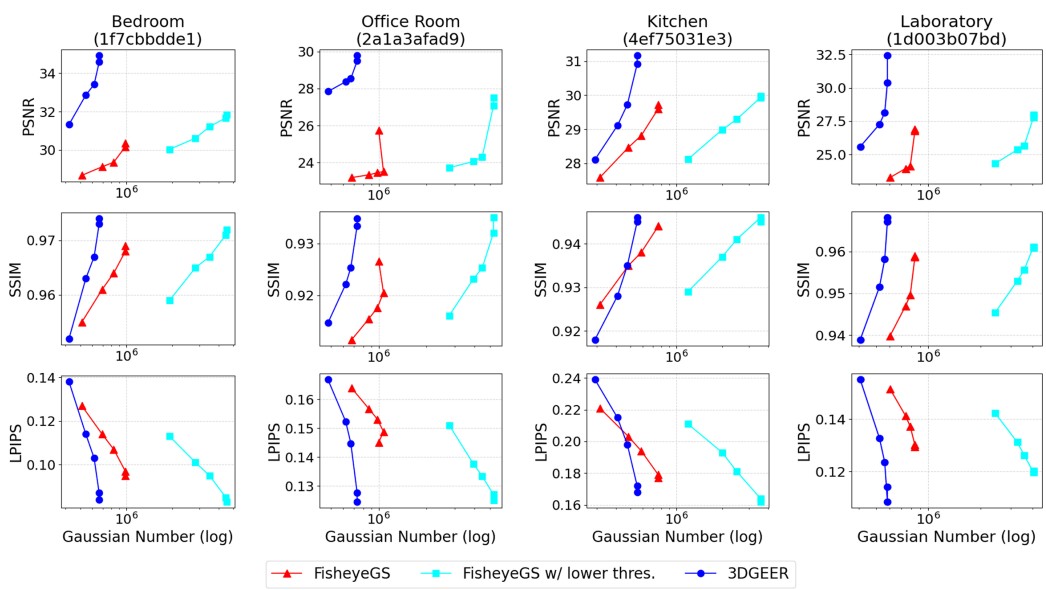

Figure K.3: **PSNR↑ SSIM↑ LPIPS↓ Trends When Scaling-Up Gaussian Counts.** Brute-force scaling fails to close the gap in projective exactness.

Table K.2: **ScanNet++ Quantitative Results by Sequence (Full FoV).** The overall results are shown in Tab. 2.

| Scene | EVER | | | FisheyeGS | | | 3DGUT | | | 3DGEER (Ours) | | |
|---|---|---|---|---|---|---|---|---|---|---|---|---|
| | PSNR | SSIM | LPIPS | PSNR | SSIM | LPIPS | PSNR | SSIM | LPIPS | PSNR | SSIM | LPIPS |
| Storage | 28.24 | 0.916 | 0.164 | 24.74 | 0.932 | 0.146 | **29.26** | 0.934 | 0.147 | 29.18 | **0.940** | **0.141** |
| Office Room | 27.80 | 0.899 | 0.169 | 25.75 | 0.927 | 0.145 | 29.37 | 0.928 | 0.145 | **29.78** | **0.935** | **0.125** |
| Bedroom | 32.60 | 0.948 | 0.131 | 31.95 | 0.970 | 0.095 | 34.65 | 0.967 | 0.112 | **34.94** | **0.974** | **0.084** |
| Kitchen | 29.77 | 0.924 | 0.213 | 29.72 | 0.944 | 0.177 | 30.78 | 0.941 | 0.196 | **31.17** | **0.946** | **0.168** |
| Laboratory | 28.96 | 0.936 | 0.156 | 26.91 | 0.959 | 0.129 | 29.12 | 0.948 | 0.151 | **32.43** | **0.968** | **0.114** |

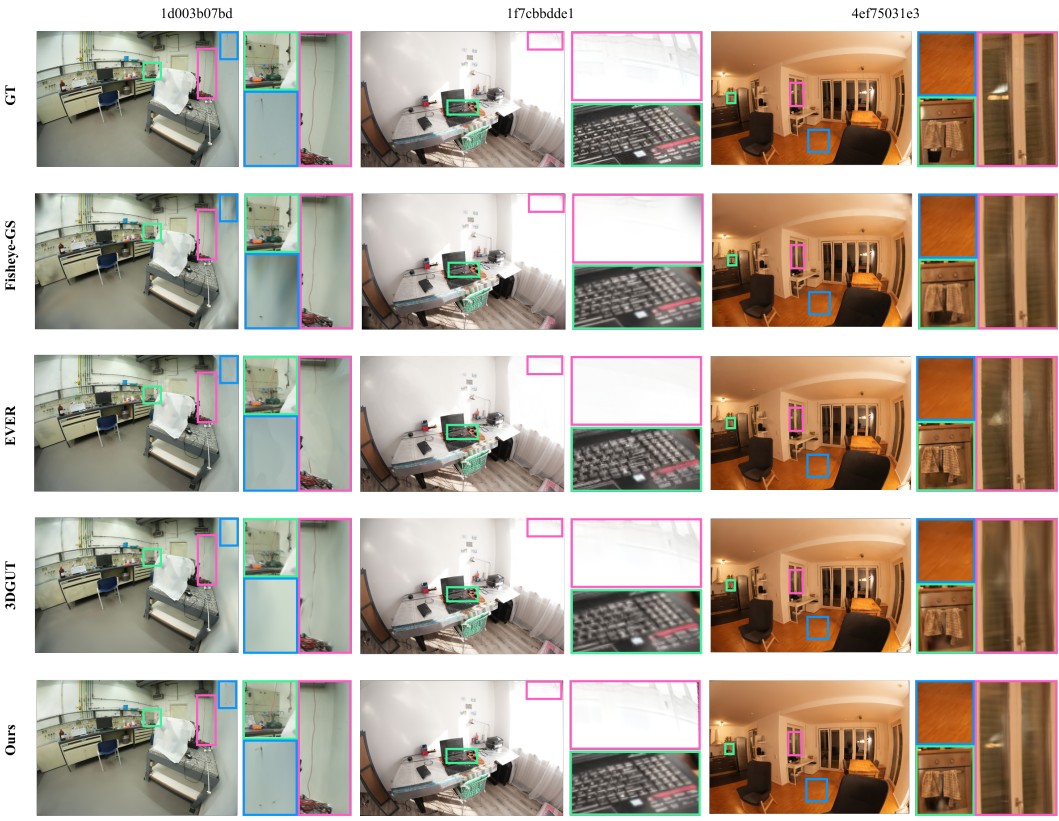

Figure K.4: **Qualitative Results on the ScanNet++ Dataset.** Larger differences are visible in the zoomed-in colored boxes, particularly in structural details and artifact regions. Our method delivers superior peripheral reconstruction compared to FisheyeGS, exhibits fewer artifacts and sharper details than EVER, and produces crisper object boundaries and textures—such as wires, keyboards, and wooden floors—when compared to 3DGUT.

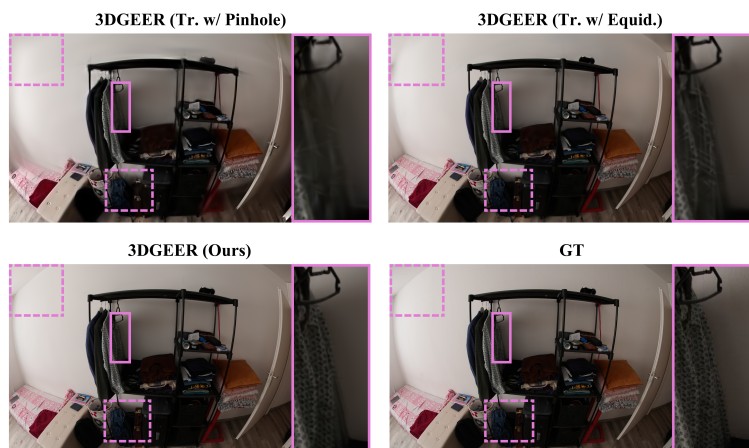

Figure K.5: **Impact of BEAP supervision.** Visual comparison on ScanNet++ shows that BEAP's balanced angular sampling produces smoother gradients and better preservation of geometric and texture details than pinhole or equidistant sampling. (Tab. K.8 shows quantitative full stats).

Table K.3: **Quantitative Results on the ScanNet++.** PSNR↑ SSIM↑ LPIPS↓ Trends When Scaling-Up Gaussian Counts. Bedroom, Office Room, Kitchen, and Laboratory in order.

| Iter | FisheyeGS | | | | FisheyeGS w/ Low Thres. | | | | 3DGEER | | | |
|---|---|---|---|---|---|---|---|---|---|---|---|---|
| | PSNR↑ | SSIM↑ | LPIPS↓ | Gau↓ | PSNR↑ | SSIM↑ | LPIPS↓ | Gau↓ | PSNR↑ | SSIM↑ | LPIPS↓ | Gau↓ |
| 4k | 28.67 | 0.955 | 0.127 | 512K | 30.03 | 0.959 | 0.113 | 1914K | 31.33 | 0.952 | 0.138 | 422K |
| 7k | 29.12 | 0.961 | 0.114 | 695K | 30.61 | 0.965 | 0.101 | 2829K | 32.87 | 0.963 | 0.114 | 543K |
| 10k | 29.36 | 0.964 | 0.107 | 827K | 31.21 | 0.967 | 0.095 | 3489K | 33.42 | 0.967 | 0.103 | 616K |
| 20k | 30.17 | 0.968 | 0.097 | 985K | 31.68 | 0.971 | 0.085 | 4467K | 34.59 | 0.973 | 0.087 | 664K |
| 30k | 30.36 | 0.969 | 0.095 | 985K | 31.84 | 0.972 | 0.083 | 4535K | 34.94 | 0.974 | 0.084 | 664K |

| Iter | FisheyeGS | | | | FisheyeGS w/ Low Thres. | | | | 3DGEER | | | |
|---|---|---|---|---|---|---|---|---|---|---|---|---|
| | PSNR↑ | SSIM↑ | LPIPS↓ | Gau↓ | PSNR↑ | SSIM↑ | LPIPS↓ | Gau↓ | PSNR↑ | SSIM↑ | LPIPS↓ | Gau↓ |
| 4k | 23.19 | 0.911 | 0.164 | 677K | 23.72 | 0.916 | 0.151 | 2778K | 27.84 | 0.915 | 0.167 | 480K |
| 7k | 23.33 | 0.916 | 0.157 | 867K | 24.06 | 0.923 | 0.138 | 3936K | 28.37 | 0.922 | 0.152 | 621K |
| 10k | 23.45 | 0.918 | 0.153 | 983K | 24.31 | 0.925 | 0.133 | 4481K | 28.54 | 0.925 | 0.145 | 668K |
| 20k | 23.51 | 0.921 | 0.149 | 1072K | 27.07 | 0.932 | 0.127 | 5272K | 29.49 | 0.933 | 0.128 | 732K |
| 30k | 25.75 | 0.927 | 0.145 | 1001K | 27.50 | 0.935 | 0.125 | 5272K | 29.78 | 0.935 | 0.125 | 732K |

| Iter | FisheyeGS | | | | FisheyeGS w/ Low Thres. | | | | 3DGEER | | | |
|---|---|---|---|---|---|---|---|---|---|---|---|---|
| | PSNR↑ | SSIM↑ | LPIPS↓ | Gau↓ | PSNR↑ | SSIM↑ | LPIPS↓ | Gau↓ | PSNR↑ | SSIM↑ | LPIPS↓ | Gau↓ |
| 4k | 27.58 | 0.926 | 0.221 | 314K | 28.12 | 0.929 | 0.211 | 1191K | 28.11 | 0.918 | 0.239 | 292K |
| 7k | 28.47 | 0.935 | 0.203 | 485K | 28.99 | 0.937 | 0.193 | 1997K | 29.11 | 0.928 | 0.215 | 410K |
| 10k | 28.82 | 0.938 | 0.194 | 588K | 29.29 | 0.941 | 0.181 | 2491K | 29.73 | 0.935 | 0.198 | 476K |
| 20k | 29.60 | 0.944 | 0.179 | 760K | 29.93 | 0.946 | 0.164 | 3563K | 30.91 | 0.945 | 0.172 | 552K |
| 30k | 29.72 | 0.944 | 0.177 | 760K | 29.97 | 0.945 | 0.162 | 3563K | 31.17 | 0.946 | 0.168 | 552K |

| Iter | FisheyeGS | | | | FisheyeGS w/ Low Thres. | | | | 3DGEER | | | |
|---|---|---|---|---|---|---|---|---|---|---|---|---|
| | PSNR↑ | SSIM↑ | LPIPS↓ | Gau↓ | PSNR↑ | SSIM↑ | LPIPS↓ | Gau↓ | PSNR↑ | SSIM↑ | LPIPS↓ | Gau↓ |
| 4k | 23.27 | 0.940 | 0.152 | 602K | 24.32 | 0.945 | 0.142 | 2429K | 25.59 | 0.939 | 0.155 | 406K |
| 7k | 23.94 | 0.947 | 0.141 | 740K | 25.36 | 0.953 | 0.131 | 3258K | 27.27 | 0.952 | 0.133 | 524K |
| 10k | 24.12 | 0.950 | 0.137 | 788K | 25.65 | 0.956 | 0.126 | 3569K | 28.14 | 0.958 | 0.124 | 562K |
| 20k | 26.76 | 0.959 | 0.130 | 838K | 27.77 | 0.961 | 0.120 | 4035K | 30.39 | 0.967 | 0.108 | 582K |
| 30k | 26.91 | 0.959 | 0.129 | 838K | 27.97 | 0.961 | 0.120 | 4035K | 32.43 | 0.968 | 0.114 | 582K |

Table K.4: **Runtime Analysis vs. Other Methods.** 3DGEER is compared with efficient splatting-based methods on MipNeRF360 and ScanNet++, with a thorough comparison in stage-wise and total runtime.

| Dataset | Method | Avg. Timings (ms) ↓ | | | | |
|---|---|---|---|---|---|---|
| | | Prep. | Dup. | Sort | Render | Total |
| MipNeRF360 | 3DGS | 0.32 | 0.27 | 0.65 | 1.40 | 2.92 |
| | 3DGEER (Ours) | 0.37 | 0.17 | 0.27 | 2.10 | 3.06 |
| ScanNet++ | FisheyeGS | 0.10 | 0.63 | 1.45 | 2.33 | 4.70 |
| | 3DGEER (Ours) | 0.13 | 0.26 | 0.59 | 2.89 | 3.98 |

Table K.5: **Runtime Analysis: Asso. Method on ScanNet++.** The efficiency of the PBF-based association is compared to the splatting-based EWA and UT with acceleration option SnugBox integrated into 3DGEER pipeline.

| 3DGEER Asso. | Prep. | Avg. Timings (ms) ↓ | | | | Avg. FPS ↑ |
|---|---|---|---|---|---|---|
| | | Dup. | Sort | Render | Total | |
| EWA | 0.13 | 1.17 | 3.27 | 12.69 | 17.84 | 56 |
| + SnugBox | 0.13 | 0.60 | 1.42 | 6.22 | 8.68 | 115 |
| UT | 0.13 | 0.71 | 1.94 | 7.97 | 11.16 | 90 |
| + SnugBox | 0.13 | 0.28 | 0.64 | 3.16 | 4.38 | 228 |
| PBF (Ours) | 0.13 | **0.26** | **0.59** | **2.89** | **3.98** | **251** |

Table K.6: **Impact of PBF-CSF Association.** The evaluation employs the artifact-sensitive LPIPS metric. To assess cross-model robustness, each variant is also tested using models trained with the other association strategies. Additionally, we report the memory cost during rendering along with the average number of Gaussian association per Tile, and the performance gap ΔLPIPS between in-method and cross-method evaluations to quantify consistency and generalization.

| Asso. Method | Gaus. ↓ per Tile | | Mem. ↓ | LPIPS ↓ | Cross-LPIPS ↓ | ΔLPIPS ↓ |
|---|---|---|---|---|---|---|
| | mean | std | (GB) | | | (1e-2) |
| EWA | 2203 | 1447 | 2.2 | 0.1250 | 0.1321 | 0.71 |
| UT | 1377 | 1058 | 1.4 | 0.1249 | 0.1302 | 0.53 |
| PBF (Ours) | **475** | **427** | **0.63** | **0.1245** | **0.1251** | **0.06** |

Table K.7: **Training-Time Transmittance Func. Replacement.** The overall results are shown in Tab. 5.

| Scene | 3DGS Splats + PBF | | | | FisheyeGS Splats + PBF | | | | 3DGEER (Ours) | | | |
|---|---|---|---|---|---|---|---|---|---|---|---|---|
| | PSNR | SSIM | LPIPS | Gaus. Num (k) | PSNR | SSIM | LPIPS | Gaus. Num (k) | PSNR | SSIM | LPIPS | Gaus. Num (k) |
| Storage | 19.27 | 0.859 | 0.173 | 1293 | 24.75 | 0.933 | 0.150 | 781 | 29.18 | 0.940 | 0.141 | 429 |
| Office Room | 20.89 | 0.848 | 0.182 | 1527 | 25.89 | 0.928 | 0.150 | 1119 | 29.78 | 0.935 | 0.125 | 732 |
| Bedroom | 27.39 | 0.923 | 0.119 | 1542 | 31.96 | 0.970 | 0.096 | 1079 | 34.94 | 0.974 | 0.084 | 664 |
| Kitchen | 25.15 | 0.890 | 0.239 | 1241 | 29.87 | 0.949 | 0.182 | 801 | 31.17 | 0.946 | 0.168 | 552 |
| Laboratory | 21.60 | 0.869 | 0.173 | 1379 | 27.03 | 0.962 | 0.131 | 823 | 32.43 | 0.968 | 0.114 | 582 |

Table K.8: **Impact of Different Supervision Space.** The overall results are shown in Tab. 6.

| Supervision Space Scene | 3DGEER w/o BEAP Central Perspective | | | 3DGEER w/o BEAP Perspective | | | 3DGEER w/o BEAP Equidistant | | | 3DGEER (Ours) BEAP | | |
|---|---|---|---|---|---|---|---|---|---|---|---|---|
| | PSNR | SSIM | LPIPS | PSNR | SSIM | LPIPS | PSNR | SSIM | LPIPS | PSNR | SSIM | LPIPS |
| Storage | 28.42 | 0.931 | 0.143 | 19.74 | 0.834 | 0.302 | 29.03 | 0.938 | 0.142 | 29.18 | 0.940 | 0.141 |
| Office Room | 28.90 | 0.927 | 0.133 | 21.12 | 0.825 | 0.291 | 29.71 | 0.932 | 0.126 | 29.78 | 0.935 | 0.125 |
| Bedroom | 33.86 | 0.965 | 0.090 | 24.38 | 0.882 | 0.240 | 34.60 | 0.973 | 0.096 | 34.94 | 0.974 | 0.084 |
| Kitchen | 30.65 | 0.941 | 0.172 | 23.23 | 0.889 | 0.338 | 31.23 | 0.941 | 0.177 | 31.17 | 0.946 | 0.168 |
| Laboratory | 27.35 | 0.951 | 0.118 | 17.08 | 0.834 | 0.329 | 30.65 | 0.956 | 0.135 | 32.43 | 0.968 | 0.114 |

Table K.9: **ZipNeRF Quantitative Results by Sequence.** Alameda, Berlin, London, and NYC in order. The overall results are shown in Tab. 3.

| Train Data | 1/8 FE | | 1/8 FE | | 1/8 FE | | 1/4 PH | | 1/4 PH | | 1/4 PH | |
|---|---|---|---|---|---|---|---|---|---|---|---|---|
| Test Data | 1/8 FE | 1/4 PH | 1/8 FE | 1/4 PH | 1/8 FE | 1/4 PH | 1/8 FE | 1/4 PH | 1/8 FE | 1/4 PH | 1/8 FE | 1/4 PH |
| Method | PSNR ↑ | | SSIM ↑ | | LPIPS ↓ | | PSNR ↑ | | SSIM ↑ | | LPIPS ↓ | |
| FisheyeGS | 20.74 | 23.64 | 0.820 | 0.802 | 0.223 | 0.277 | 17.39 | 23.74 | 0.736 | 0.839 | 0.265 | 0.240 |
| EVER | 22.46 | 23.06 | 0.852 | 0.777 | 0.144 | 0.274 | 20.06 | 22.83 | 0.789 | 0.794 | 0.213 | 0.250 |
| 3DGUT | 21.78 | 22.39 | 0.841 | 0.718 | 0.193 | 0.368 | 17.03 | 22.58 | 0.588 | 0.765 | 0.329 | 0.311 |
| Ours | 24.60 | 25.00 | 0.857 | 0.839 | 0.154 | 0.241 | 21.49 | 25.21 | 0.792 | 0.807 | 0.232 | 0.277 |

| Train Data | 1/8 FE | | 1/8 FE | | 1/8 FE | | 1/4 PH | | 1/4 PH | | 1/4 PH | |
|---|---|---|---|---|---|---|---|---|---|---|---|---|
| Test Data | 1/8 FE | 1/4 PH | 1/8 FE | 1/4 PH | 1/8 FE | 1/4 PH | 1/8 FE | 1/4 PH | 1/8 FE | 1/4 PH | 1/8 FE | 1/4 PH |
| Method | PSNR ↑ | | SSIM ↑ | | LPIPS ↓ | | PSNR ↑ | | SSIM ↑ | | LPIPS ↓ | |
| FisheyeGS | 24.25 | 28.25 | 0.895 | 0.924 | 0.206 | 0.209 | 20.08 | 27.86 | 0.848 | 0.929 | 0.233 | 0.194 |
| EVER | 25.26 | 26.97 | 0.912 | 0.907 | 0.155 | 0.221 | 23.18 | 27.72 | 0.877 | 0.913 | 0.196 | 0.212 |
| 3DGUT | 25.37 | 27.02 | 0.904 | 0.876 | 0.197 | 0.288 | 18.57 | 27.43 | 0.730 | 0.898 | 0.297 | 0.249 |
| Ours | 25.51 | 29.15 | 0.916 | 0.925 | 0.166 | 0.194 | 23.58 | 29.51 | 0.878 | 0.915 | 0.219 | 0.228 |

| Train Data | 1/8 FE | | 1/8 FE | | 1/8 FE | | 1/4 PH | | 1/4 PH | | 1/4 PH | |
|---|---|---|---|---|---|---|---|---|---|---|---|---|
| Test Data | 1/8 FE | 1/4 PH | 1/8 FE | 1/4 PH | 1/8 FE | 1/4 PH | 1/8 FE | 1/4 PH | 1/8 FE | 1/4 PH | 1/8 FE | 1/4 PH |
| Method | PSNR ↑ | | SSIM ↑ | | LPIPS ↓ | | PSNR ↑ | | SSIM ↑ | | LPIPS ↓ | |
| FisheyeGS | 23.57 | 26.65 | 0.829 | 0.861 | 0.253 | 0.254 | 20.83 | 27.33 | 0.768 | 0.883 | 0.288 | 0.229 |
| EVER | 24.59 | 26.83 | 0.823 | 0.859 | 0.218 | 0.223 | 23.14 | 26.98 | 0.793 | 0.860 | 0.268 | 0.235 |
| 3DGUT | 25.61 | 26.25 | 0.854 | 0.803 | 0.221 | 0.336 | 19.82 | 26.41 | 0.669 | 0.833 | 0.313 | 0.291 |
| Ours | 26.60 | 27.84 | 0.895 | 0.882 | 0.142 | 0.230 | 23.95 | 27.15 | 0.840 | 0.846 | 0.217 | 0.289 |

| Train Data | 1/8 FE | | 1/8 FE | | 1/8 FE | | 1/4 PH | | 1/4 PH | | 1/4 PH | |
|---|---|---|---|---|---|---|---|---|---|---|---|---|
| Test Data | 1/8 FE | 1/4 PH | 1/8 FE | 1/4 PH | 1/8 FE | 1/4 PH | 1/8 FE | 1/4 PH | 1/8 FE | 1/4 PH | 1/8 FE | 1/4 PH |
| Method | PSNR ↑ | | SSIM ↑ | | LPIPS ↓ | | PSNR ↑ | | SSIM ↑ | | LPIPS ↓ | |
| FisheyeGS | 24.15 | 27.21 | 0.890 | 0.885 | 0.162 | 0.215 | 19.41 | 27.50 | 0.811 | 0.907 | 0.203 | 0.183 |
| EVER | 28.39 | 27.71 | 0.931 | 0.860 | 0.095 | 0.228 | 25.22 | 28.27 | 0.879 | 0.883 | 0.154 | 0.192 |
| 3DGUT | 26.31 | 26.69 | 0.917 | 0.817 | 0.121 | 0.302 | 19.00 | 27.42 | 0.659 | 0.866 | 0.263 | 0.228 |
| Ours | 28.26 | 28.49 | 0.922 | 0.906 | 0.097 | 0.190 | 24.52 | 28.55 | 0.874 | 0.885 | 0.168 | 0.220 |

