# OpenReview forum: "3DGEER: 3D Gaussian Rendering Made Exact and Efficient for Generic Cameras"
_ICLR.cc/2026/Conference — ICLR 2026 Poster_

### Official Review · Reviewer_62Nx · 2025-10-26

**Soundness:** 4
**Presentation:** 4
**Contribution:** 4
**Rating:** 10
**Confidence:** 5

**Summary:**

This paper investigates how Gaussians are projected within the 3D Gaussian Splatting framework and addresses the limitations of existing projection methods for general cameras, particularly those with large fields of view (FoVs) where standard approaches often fail. Specifically, it analyzes an approximation commonly used in the projection process and demonstrates that this approximation can negatively affect the accuracy of Gaussian projection under large FoVs.

To achieve general and accurate Gaussian projection, the paper makes three main contributions. First, it shows that the transmittance of a Gaussian depends solely on the distance between a ray and the Gaussian’s center. Second, it introduces a more precise yet computationally efficient formulation for determining Gaussian bounds. Third, it proposes Bipolar Equiangular Projection (BEAP), a novel technique for enhanced ray sampling that provides more uniform coverage and improves projection efficiency.

Through experiments on diverse datasets and camera configurations, the proposed method demonstrates notable improvements in representation quality and significantly reduces the number of Gaussians required. In contrast, prior methods relied on a much larger number of Gaussians to compensate for distortions in large FoVs.

**Strengths:**

**Clear and well-structured presentation**: The paper is well-written and easy to follow. It clearly articulates the problems addressed, the proposed solutions, and their significance, supported by clear and informative figures.

**Solid theoretical foundation**: The paper provides comprehensive mathematical formulations necessary to understand the proposed method, with additional derivations and supporting details presented in the appendix. It is also noteworthy that the paper explores alternative angular representations to overcome the limitations of conventional tan-based approaches (Appendix D.2).

**Strong quantitative and qualitative results**: The proposed method demonstrates consistent performance improvements across diverse datasets and camera types, including both fisheye and pinhole cameras.

**Weaknesses:**

The contribution of the transmittance equation (Eq. 5) to the overall method could be articulated more clearly. In particular, the benefits of this formulation for general camera models are not fully evident, and the paper could better explain how this component integrates with and enhances the proposed framework.

**Questions:**

1. Very minor, but in Fig. K.2, the “laboratory” and “office” scenes appear to be mixed.

---

> ### Author Response · Authors · 2025-11-23
> **Response**
>
> We sincerely appreciate your positive and insightful review. Please find our detailed responses below.
>
> ---
> ##### **W1: How projective-exact transmittance equation enhances the proposed framework.**
>
> Thank you for the insightful question. Below we clarify how the projective-exact transmittance equation directly strengthens the proposed framework, supported by the new controlled ablations added in *Sec. 4.4*
>
> To directly measure the effect of the projective-exact transmittance, we replace only the transmittance computation at test time with the commonly used local-affine splat approximation, while keeping the training pipeline fully projective-exact.
> As shown in *Fig. 8*, this single substitution already breaks the consistency between exact association and approximate splatting, causing mismatched culling and gridline-like artifacts.
>
>
> We further replace the transmittance with splatting during both training and testing, again keeping association projective-exact. *Tab. 5* (full stats in *Tab. K.7*) shows that, in this mismatched setting, the optimizer and densification attempt to compensate for the cross-view inconsistency introduced by the approximate transmittance. However, this compensation requires 1.6–2.4× more Gaussians, and even then residual perceptual gaps remain.
>
>
> Across both ablations, the results show that the projective-exact transmittance equation is the key component that enforces correct projective geometry, ensures consistent culling behavior, stabilizes optimization, and prevents the model from over-densifying.
>
> | Transmittance  | Full PSNR ↑ | Cen. PSNR ↑ | Per. PSNR ↑ | # Gaussians (K) ↓  |
> |-----------------------|------|-------------|-------------|----------------|
> | 3DGS Splats        |  22.86  | 27.16       | 18.56       |  1396 (2.4x)  |
> | FisheyeGS Splats  |  27.90  | 32.49       | 23.36       |  921 (1.6x)   |
> | Proj.-Exact (Ours) |  31.50  | 32.64       | 28.94       |  592          |
>
>
> ---
> ##### **W2: Figure K.2 revision.**
>
> A: Thank you for catching this. We have corrected the image arrangement in *Figure K.2*.

---

### Official Review · Reviewer_SpuR · 2025-10-27

**Soundness:** 3
**Presentation:** 3
**Contribution:** 3
**Rating:** 6
**Confidence:** 4

**Summary:**

This paper presents a new 3D Gaussian renderer designed for large field-of-view cameras, such as fish-eye cameras. It derives a closed-form expression for integrating the Gaussian density along a ray. To enhance efficiency, the paper introduces the Particle Bounding Frustum (PBF), which provides a tight bounding box to cull unnecessary ray-Gaussian intersections.

Unlike pinhole cameras, where rays are typically sampled uniformly across a 2D plane, this paper proposes sampling rays uniformly from a sphere using Equiangular Projection. This approach allows for interpolation of the rays to produce the output image, further improving quality.

Experiments conducted on both pinhole and fish-eye datasets demonstrate that the 3D Gaussian Efficient Ray Renderer (3DGEER) outperforms previous methods across all metrics. Additionally, it runs five times faster than existing projective exact ray-based baselines and generalizes well to wider fields of view that were not seen during training.

**Strengths:**

1. **Strong results.** The experimental results are impressive, achieving state-of-the-art performance across many large field-of-view (FOV) datasets. Additionally, there are significant improvements in memory efficiency and rendering speed compared to 3DGRT and EVER.

2. **Good presentation.** The paper is well-structured and easy to read. The derivation of the formula is generally clear. However, some sections, such as the gradient flow in Figure 2, could be streamlined or moved to the appendix, as they do not seem essential for all readers.

**Weaknesses:**

**Limited discussion to existing work**.  The paper provides limited  discussion regarding existing work. While there are several derivations of the proposed techniques, including the closed-form expression for integrating Gaussian functions and the tight bounding box, these derivations have been adopted in prior research without clear reference in this paper.

For the exact projective geometry, Equation 5 in this paper is identical to Equation 15 in HTGS [1].  Similarly, the equations for calculating the tight bounding box (Equations 7 through 10) appear to be equivalent to Equations 8 through 10 in both 2DGS [2] (with the substitutions x = tan(θ) and y = tan(φ)). Both of these derivations originate from source [3]. However, these papers are not explicitly cited, and a more thorough discussion regarding their connections should be included. (The only reference seems to be in Section 2.2 (lines 139-142), which is not entirely accurate since it does not use BVH. )

Furthermore, due to the similarities between these works, it is unclear how this new formula can eliminate the screen-space constraints present in HTGS and 2DGS that limit their field of view, as claimed in lines 285-286. This point requires additional justification.

[1] Efficient Perspective-Correct 3D Gaussian Splatting Using Hybrid Transparency. Eurographics 2025.
[2] 2D Gaussian Splatting for Geometrically Accurate Radiance Fields. SIGGRAPH'24.
[3] A hardware architecture for surface splatting. ACM TOG 2007.

**Questions:**

See W1 in weakness.

---

> ### Author Response · Authors · 2025-11-23
> **Response**
>
> We truly appreciate your valuable comments. In the following, we provide responses to the concerns.
>
> ##### **W1: Limited discussion of existing work.**
> A: Thank you for the insightful question. We have added the following discussion in *Sec. D.3*.
>
> When substituting the perspective projection
> (x=tan$\theta$, y=tan$\phi$) into our formulation, the resulting expressions match those in HTGS [1] and 2DGS [2]. This is expected -- as it is a ***special case of our general association model*** under perspective projection.
>
> The key difference lies in the assumptions underlying the derivations. HTGS [1] and 2DGS [2] work directly in projective screen space, where the tangent plane is implicitly assumed to be planar and parallel to the z-axis. This assumption is valid only for perspective cameras and cannot extend to fisheye or omnidirectional systems, where the corresponding surface shooting from their screen space is curved.
>
> In contrast, our derivation starts from a camera-agnostic geometric formulation directly parameterizing the angular limits of the bounds. This yields a unified expression applicable to arbitrary generic camera models, with perspective projection simply emerging as one special case. Additionally, *Sec. D.2* includes implementation details required for making PBF reliable under large-FoV and omnidirectional cameras—scenarios not addressed in prior work.
>
> We’ve added *Sec. D.3*  to discuss these works, clarifying these derivational relationships and more explicitly state the conditions under which prior formulas coincide with ours.
>
> *[1] Efficient Perspective-Correct 3D Gaussian Splatting Using Hybrid Transparency. Eurographics 2025.*
>
> *[2] 2D Gaussian Splatting for Geometrically Accurate Radiance Fields. SIGGRAPH'24.*
>
> ---
> ##### **W2: Correctly cite 2DGS.**
> A: Thank you for catching this.
>
> The confusion was caused by an unintended sentence ordering (sentence 2 and 3 were swapped) in *Sec. 2.2*, which may have implied that both EVER and 2DGS rely on BVH traversal. Our intended meaning was: only EVER uses BVH, while 2DGS should not be included.
>
> We have corrected the phrasing in *Sec. 2.2* (blue colored).

---

> ### Comment · Reviewer_SpuR · 2025-11-26
> **Response by the reviewer**
>
> Thank you for the detailed response and for clarifying your perspective on the sampling strategy. I understand the distinction you're making: your method samples rays based on spherical coordinates (θ, φ), whereas prior work often parameterizes them by pixel coordinates (x, y) on a virtual tangent plane.
>
> However, my concern about the framing of this contribution remains. While the sampling parameterization is indeed different, the subsequent computational pipeline for processing these rays appears to be the same.
>
> Therefore, framing these prior methods as a 'special case' of your formulation feels like an **overstatement of the contribution**. A 'special case' typically implies that the more general framework can be directly simplified to the older one under specific constraints. Here, it seems more accurate to describe them as alternative (though closely related) formulations for solving the same underlying rendering problem. The choice between them is dictated by the desired sampling geometry (e.g., spherical vs. planar) rather than a fundamental change in the rendering equation itself.
>
> I strongly believe the paper would be improved by addressing this comparison more directly and precisely in the main text, rather than omitting it. Hiding this discussion could be perceived by readers as an attempt to obscure the similarities.
>
> My recommendations would be:
>
> 1. **Reframe the contribution**. In the main paper, please rephrase the claim and add reference. Instead of entirely omitting them or calling prior work a 'special case,' you could present your method as a more flexible formulation that naturally handles spherical or 360° scenarios, and then explicitly discuss its relationship to planar-based sampling methods.
> 2. **Include the Comparison in the Main Text**: A concise paragraph in the methodology or related work section that acknowledges the similarities in the rendering pipeline but highlights the advantages of your θ, φ parameterization would significantly strengthen the paper. Please also include HTGS in the Table.1.
> 3. **Revise the title**: reduce your claim about "Generic Cameras" since your approach is specifically tailored for fisheye cameras. "Fisheye cameras" could be more appropriate.
>
> I am currently uncertain about acceptance unless these points are well addressed.

---

> > ### Author Response · Authors · 2025-11-27
> > **Follow-up Response (Part II)**
> >
> > ##### **On the framing of contribution:**
> >
> > Concretely, HTGS/2DGS start from a screen-space or tangent-plane parameterization and therefore rely on assumptions that hold only for perspective (pinhole) cameras (e.g., geometric identities that arise when rays are implicitly tied to a planar, z-parallel image plane). In contrast, our derivation begins from a camera-agnostic angular parameterization in 3D and directly parameterizes Camera Sub-Frustums (CSFs) and Particle Bounding Frustums (PBFs). These different starting assumptions change the known quantities and constraints available during derivation, and thus the mathematical conditions under which closed-form bounds and culling guarantees remain valid for generic cameras.
> >
> > We’d like to emphasize that mathematics in itself is a general tool, but deriving these bounds under weaker assumptions (i.e., **without relying on screen-space structure**) and validating them with comprehensive experiments across both pinhole and wide-FOV camera families constitute an essential part of our contribution.
> >
> > Moreover, the downstream computational pipeline is affected by this choice of parameterization. Gaussian association, tiling/duplication, and sorting depend on how Gaussians map to tiles/bins — with an angular (CSF-based) parameterization **we obtain a different tiling/bucketing structure in angular domain, rather than tiling in a pinhole screen space.** Practically, this matters for large-FoV scenes: under generic cameras a Gaussian’s bounding can become partially “behind” the camera in screen coordinates (see Fig. D.1), which breaks assumptions used in screen-space as well (since **tan value can provide a virtually mirrored bound, which is incorrect**, see dashed line in Fig. D.1). To handle these cases we introduce mirror parameters into the association (inspired by CMEI [1]) and compute the correct angles from there, which is not required in perspective-only derivations. The Gaussian center is also involved to sort and correct the bounds.
> >
> > For these reasons, we view the perspective/pinhole formulations as a camera-model special case of our more general angular-domain expression rather than as an alternative parameterization derived under the same assumptions and with an identical computational pipeline. Section D in the Appendix (D.1 and D.2 in the **original** manuscript, with an extended D.3 in the revision) is intended to make this distinction transparent, showing both the algebraic correspondence and the substantive differences in assumptions, implementation behavior, and applicability.
> >
> > As suggested, we’ve replaced the wording **“special case”** with **“pinhole alternative”** and also added HTGS in *Tab.1*. We have pointed to this section D in our original main text (*Section 3.2*) to ensure readers can easily access this comparison. We have also added this pointer to section 2.2.
> >
> >
> > [1] Mei, Christopher, and Patrick Rives. "Single view point omnidirectional camera calibration from planar grids." Proceedings 2007 IEEE International Conference on Robotics and Automation. IEEE, 2007.

---

> ### Author Response · Authors · 2025-11-27
> **Follow-up Response (Part I)**
>
> We sincerely appreciate your follow-up comments and the opportunity to clarify our position.
>
> ---
> ##### **Regarding the concerns about “omitting” or “hiding”**
> We respectfully disagree with the wording suggesting that prior work was “not explicitly cited”, or that we “omit” or “hide” relevant discussion. To avoid any misunderstanding: both HTGS[1] and 2DGS[2] were already explicitly cited in the ***original*** submission (*Sec. 2.2*, *Sec 3.2* with a pointer to *Appendix D*, see manuscript revision history 09.24). Following your earlier suggestion, our revision further expands the discussion in *Appendix D* across these methods and fixes the misunderstanding caused by swapped sentences. Our intention has never been to "omit" or "hide" related work.
>
> Due to ICLR’s strict page limits, the extended discussion of our PBF was placed in the *Appendix D* together with its geometric derivation and the other implementation details—this is a standard and intended use of the Appendix. We have further expanded the main-text as requested and as space permits.
>
> We kindly ask that the evaluation focus on the factual content of the manuscript rather than implying intentional omission, as such phrasing may inadvertently misrepresent the actual citations and discussions we have provided.
>
> [1] Efficient Perspective-Correct 3D Gaussian Splatting Using Hybrid Transparency. Eurographics 2025.
>
> [2] 2D Gaussian Splatting for Geometrically Accurate Radiance Fields. SIGGRAPH'24.
>
>
> ---
> ##### **On the terminology “generic cameras”**
> The term “generic camera” has long been used in the imaging and geometric vision literature to describe unified representations encompassing both perspective and a variety of fisheye projection models (CMEI [1], KB [2], equidistant, etc.). **Our method explicitly handles pinhole as well as these fisheye families within a single formulation**, and in this widely accepted sense, “generic” accurately describes our camera model coverage. We further made the above “generic” definition explicit in *Section 1*.
>
> [1] Mei, Christopher, and Patrick Rives. "Single view point omnidirectional camera calibration from planar grids." Proceedings 2007 IEEE International Conference on Robotics and Automation. IEEE, 2007.
>
> [2] Kannala, Juho, and Sami S. Brandt. "A generic camera model and calibration method for conventional, wide-angle, and fish-eye lenses." IEEE transactions on pattern analysis and machine intelligence 28.8 (2006): 1335-1340.

---

### Official Review · Reviewer_uR7S · 2025-10-29

**Soundness:** 3
**Presentation:** 3
**Contribution:** 3
**Rating:** 6
**Confidence:** 3

**Summary:**

The paper proposes, 3DGEER, a ray-based Gaussian rendering framework that achieves projective exactness with a closed-form integral of anisotropic Gaussians along rays via a canonical transform, eliminating projection linearization error. It introduces Particle Bounding Frustum (PBF) for exact, BVH-free ray–particle association and Bipolar Equiangular Projection (BEAP) for unified arbitrary-FoV sampling that accelerates association and improves quality. Experiments indicate that 3DGEER high FPS while preserving projective exactness across camera models, achieving consistent quality gains over prior exact-ray or splatting baselines.

**Strengths:**

- The paper proposes a projectively exact closed-form ray integral and an exact, BVH-free Particle Bounding Frustum (PBF) for ray–particle association, improving both accuracy and efficiency.
- The paper proposes BEAP, which uniformly samples rays in angular space, to improve generalization to large FoVs and improve efficiency and rendering quality.
- The method achieves SOTA performance compared to the baselines in both photometric metrics and FPS across camera models and generalizes to wider FoVs than those seen in training data

**Weaknesses:**

- Missing ablation for projective exactness: The paper does not include a controlled ablation that replaces the projective closed-form integral with 2D projective-space approximation (that is used in baselines) within the same pipeline. Such a study is crucial to learn how much improvement is brought specifically from the projective exact formulation (separate from PBF/BEAP)
- Missing ablation for BEAP: The paper omits a controlled BEAP on/off study within the same pipeline and comparisons to alternative projections (e.g., perspective, equidistant, equal-area). A clear ablation should report quality and efficiency with/without BEAP, stratified by image region (full FoV, center vs. edge/periphery), to identify how much quantitative improvement is brought by BEAP (separate from the other 2 contributions).
- In table 4, the runtime for some baselines are evaluated with different GPU models. It would be more clear if everything could be evaluated with a same GPU model.

**Questions:**

See weaknesses. I suggest to add the missing ablations to better understand how much improvement each contribution brings separately, and re-run some of the baselines on a consistent GPU for a coherent runtime comparison.

---

> ### Author Response · Authors · 2025-11-23
> **Response (Part I)**
>
> We truly appreciate your valuable comments. In the following, we provide responses to the concerns.
>
> ---
> ##### **W1: Missing ablation for projective exactness.**
> A: Thank you for raising this point — it helped us strengthen the paper. We have added controlled ablations to Sec. 4.4 that isolate the contribution of our projective-exact transmittance formulation from PBF and BEAP.
>
> **Rendering-time approximation.**
> We replace only the transmittance computation with the commonly used local-affine splats at test time, while training still uses our exact formulation. As shown in Fig. 8 (see our revised manuscript), this mismatch between exact association and approximated splatting produces incorrect culling and visible gridline artifacts. This experiment mirrors and explains the gridline issue reported in 3DGUT [1], but from the opposite mismatch direction.
>
> **Training-time approximation.**
> We further replace the transmittance with splatting during both training and testing while keeping projective-exact association. As shown in the table below (Tab. 5 / K.7 shows full stats in our revised manuscript), the optimizer and densification partially compensate for the larger cross-view inconsistency, but this compensation comes at the cost of 1.6–2.4x more Gaussians. Even with this increase, residual perceptual gaps remain, confirming that linear local-affine splats cannot fully correct the projective inconsistency.
>
> | Transmittance Meth.  | Central PSNR ↑ | Peripheral PSNR ↑ | # Gaussians  |
> |-----------------------|-------------|-------------|----------------|
> | 3DGS Splats           | 27.16       | 18.56       |  1396 K (2.4x)  |
> | FisheyeGS Splats      | 32.49       | 23.36       | 921 K (1.6x)   |
> | Proj.-Exact (Ours)    | 32.64       | 28.94       |  592 K          |
>
> Together, these ablations demonstrate that projective-exact transmittance is crucial for avoiding mismatch-induced artifacts and for maintaining compact Gaussian counts.
>
> *[1] Wu, Qi, et al. "3dgut: Enabling distorted cameras and secondary rays in gaussian splatting." Proceedings of the Computer Vision and Pattern Recognition Conference. 2025.*

---

> ### Author Response · Authors · 2025-11-23
> **Response (Part II)**
>
> ##### **W2: Missing ablation for BEAP.**
>
> A: Thank you for raising this point — it helped us strengthen the paper.
>
> We have added the requested BEAP on/off study to *Sec. 4.5*. In this study, we keep the projective-exact integral and PBF components fixed, and vary only the ray-sampling distribution used for supervision (pinhole, equidistant, and BEAP).
>
> As shown in the table below (*Tab. 6 / K.8* shows full stats and *Fig. K.5* shows visuals), BEAP consistently outperforms all alternatives across all metrics on ScanNet++. Its more uniform angular sampling produces smoother gradients, improves optimization stability, and yields more accurate recovery of fine details. The effect is especially pronounced compared with large-FoV pinhole cameras due to its highly non-uniform angular coverage; the peripheral region dominates the area but contains less informative signal (see *Fig. K.2*). As a result, simply expanding the FoV can even hurt performance (rows 1–2), whereas BEAP maintains balanced supervision when expanding to larger FoV.
>
> | Supervision           | Full PSNR ↑ | Cen. PSNR ↑ | Per. PSNR ↑ |
> |-----------------------|-------------|-------------|-------------|
> | Central Persp.        | 29.84       | 32.21       |  26.23       |
> | Perspective           | 21.11       | 21.32       |  20.46       |
> | Equidistant           | 31.05       | 32.21       |  28.56       |
> | Central BEAP (see *Tab. 2*)           | 29.93       | **32.86**       |  26.64       |
> | BEAP (Ours)           | **31.50**       | **32.64**       |  **28.94**       |
>
> ---
> ##### **W3:  Table 4 GPU Consistency.**
> A: Thank you for pointing out the GPU differences in Table 4.
>
> For baselines such as 3DGRT [1] (SIGGRAPH 2024) and 3DGUT [2] (CVPR 2025 Oral), we used their officially reported runtimes in *Table 4* († on RTX 6000 Ada and ‡ on RTX 5090). These systems rely on heavily re-engineered hybrid (Slang+CUDA) rendering kernels and no longer share the same CUDA rasterization pipeline as classical 3DGS style. As a result, re-running their full pipelines on the same GPU (e.g., RTX 4090) could still introduce unfair comparisons w.r.t the method itself.
>
> Importantly, RTX 6000 Ada is broadly comparable to RTX 4090—both share the Ada Lovelace architecture and are built on the same AD102 GPU core—while RTX 5090 is strictly faster. Using the authors’ reported runtimes is therefore a conservative choice that does not overstate our relative performance. Meanwhile, for further fairness—and to highlight the runtime advantages of our ray–particle association—*Section 4.3* provides ablations conducted entirely within our renderer (including EWA and UT), all evaluated on the same GPU (RTX 4090).
>
> *[1] Moenne-Loccoz, Nicolas, et al. "3d gaussian ray tracing: Fast tracing of particle scenes." ACM Transactions on Graphics (TOG) 43.6 (2024): 1-19.*
>
> *[2] Wu, Qi, et al. "3dgut: Enabling distorted cameras and secondary rays in gaussian splatting." Proceedings of the Computer Vision and Pattern Recognition Conference. 2025.*

---

### Official Review · Reviewer_H5DA · 2025-10-31

**Soundness:** 3
**Presentation:** 3
**Contribution:** 3
**Rating:** 6
**Confidence:** 4

**Summary:**

This paper presents a theoretically grounded and practically efficient framework for 3D Gaussian rendering under arbitrary camera models. It derives an analytical ray–Gaussian integral by transforming anisotropic Gaussians into a canonical isotropic space, yielding an exact closed-form transmittance that unifies differentiable rendering and projective geometry without approximation. A key innovation is the Particle Bounding Frustum, which computes exact angular bounds from each Gaussian’s covariance to perform direct frustum–ray association, replacing BVH traversal while preserving geometric correctness. The method further introduces a Bipolar Equiangular Projection that provides uniform ray parameterization and improves stability across wide-FoV distortions. Experimental results demonstrate the significant performance improvement upon existing works.

**Strengths:**

1. Introduces an efficient PBF-based association strategy that achieves both geometric exactness and near real-time rendering, effectively bridging the gap between ray-tracing accuracy and splatting efficiency.
2. The proposed framework generalizes naturally across pinhole, fisheye, and omnidirectional cameras through its unified angular-space formulation and the Bipolar Equiangular Projection.
3. Demonstrates strong experimental performance and consistent generalization across diverse camera models, datasets, and fields of view, with results that are comprehensive and sufficiently support the paper’s claims.

**Weaknesses:**

1. An ablation study on the key components is missing.

2. A more detailed discussion of the limitations is needed, for example explaining how sensitive the framework is to ordering effects as discussed in related works.

**Questions:**

1. The paper reports achieving higher quality with far fewer Gaussians, but it is unclear whether this reduction stems from better ray–particle association, regularization effects, or implementation details.

2. How does the method scale to multi-million Gaussians in large scenes? Without BVH traversal, are there memory or parallelization bottlenecks at high particle counts?

3. On standard pinhole datasets, does rendering quality keep improving with more Gaussians,

4. Beyond numerical stability, what are the key algorithmic differences between 3DGEER and GOF in ray–Gaussian intersection and gradient computation?

---

> ### Author Response · Authors · 2025-11-23
> **Response (Part I)**
>
> We truly appreciate your valuable comments. In the following, we provide responses to the concerns.
>
> ##### **W1: Missing Ablation.**
>
> A: Thank you for raising this point — it helped us make the paper more complete and more concrete. We have added the requested ablations to *Sec. 4.4* and *Sec. 4.5*, which also provide additional insights into our method.
>
> In *Sec. 4.4*, we isolate and ablate the transmittance computation.
>
> **Finding 1**: If either transmittance or association is not projectively exact, gridline artifacts can easily emerge.
>
> **Finding 2**: Linear-approximation–based splats must compensate for larger cross-view inconsistency errors, which induces larger gradients and consequently heavier densification.
>
> In *Sec. 4.5*, we isolate and ablate different forms of training-time supervision. The main conclusion is that sometimes **a well-balanced distribution of supervisory information matters even more than simply expanding the FoV**, and our BEAP formulation outperforms both the pinhole and equidistant designs.
>
> ---
> ##### **W2: A more detailed discussion of the limitations.**
>
> A: Thank you for requesting a more detailed discussion of the limitations.
>
> Our current formulation integrates each Gaussian independently and therefore does not explicitly resolve self-occlusion; ordering is handled using depth-based sorting. As shown qualitatively in the supplementary videos, this may lead to occasional popping artifacts. To our knowledge, EVER [1] (ICCV 2025 Oral) is the only method that analytically resolves ordering and self-occlusion by approximating each Gaussian as a constant-density ellipsoid, though this results in a substantial drop in rendering speed.
>
> Importantly, prior ray-tracing–based Gaussian renderers – e.g., Fuzzy Metaballs [2] (ECCV 2022), and subsequent works such as 3DGRT [3] (SIGGRAPH 2024) and 3DGUT [4] (CVPR 2025 Oral) – rely on the maximum-response assumption for ray–Gaussian interaction. This assumption inherently ignores ordering and self-occlusion, yet has been shown to introduce negligible error in practice (see [5]'s Fig. 9-11 / Sec. 7.1)  regardless of the number of Gaussians. This explains that our depth-based ordering still can achieve the same practical accuracy as existing ray-tracing–based Gaussian renderers.
>
> We’ve included this **limitation** along with a **future work** discussion in *Appendix A*.
>
> *[1] Mai, Alexander, et al. "Ever: Exact volumetric ellipsoid rendering for real-time view synthesis." Proceedings of the IEEE/CVF International Conference on Computer Vision. 2025.*
>
> *[2] Keselman, Leonid, and Martial Hebert. "Approximate differentiable rendering with algebraic surfaces." European Conference on Computer Vision. Cham: Springer Nature Switzerland, 2022.*
>
> *[3] Moenne-Loccoz, Nicolas, et al. "3d gaussian ray tracing: Fast tracing of particle scenes." ACM Transactions on Graphics (TOG) 43.6 (2024): 1-19.*
>
> *[4] Wu, Qi, et al. "3dgut: Enabling distorted cameras and secondary rays in gaussian splatting." Proceedings of the Computer Vision and Pattern Recognition Conference. 2025.*
>
> *[5] Celarek, Adam, et al. "Does 3D Gaussian Splatting Need Accurate Volumetric Rendering?." Computer Graphics Forum. 2025.*
>
> ---
> ##### **Q1:  Where the Gaussian number reduction stems**
> A: Thank you for this insightful question.
>
> The reduction in Gaussian count primarily stems from **improved cross-view consistency**, enabled by our exact projective geometry. This substantially reduces cross-view approximation error—particularly under large-FoV and non-pinhole projections—so gradients remain stable and less exceed the densification threshold.
> To isolate this projective approximation error, we added an ablation (*Section 4.4*) where we replace our projective-exact transmittance formulation with splatting-based approximations during training, keeping other components fixed. As expected, the latter introduces larger rendering error, leading to noisier gradients and more aggressive densification. The below comparison is averaged across 5 ScanNet++ scenes (*Table K.7* shows full stats):
> | Transmittance Method       | PSNR ↑ | # Gaussians (K) ↓  |
> |----------------------------|--------|----------------|
> | 3DGS Splats                | 22.86  | 1396 (2.4x)  |
> | FisheyeGS Splats           | 27.90  | 921 (1.6x)   |
> | Projective-Exact (Ours)    | 31.50  | 592          |

---

> ### Author Response · Authors · 2025-11-23
> **Response (Part II)**
>
> ##### **Q2: How does the method scale to multi-million Gaussians?**
>
> A: Thank you for raising this question on large-scale scenes.
>
> Scalability to multi-million Gaussians is orthogonal to our contribution, which focuses on exact projective geometry for association and rendering. Our formulation is fully compatible with existing acceleration strategies for large-scale Gaussian representations, including hierarchical Gaussians [1] (TOG 2024), block-wise scene partitioning [2] (ECCV 2024), and Gaussian distributed training systems [3] (ICLR 2025 Oral) developed in recent 3DGS literature. These techniques operate at the level of scene representation or visibility filtering, and therefore can be applied directly on top of our projective-exact Gaussian rendering method.
>
> Importantly, our method does not introduce additional memory complexity beyond the per-Gaussian parameters already used in existing frameworks, and it parallelizes identically to standard 3DGS rasterization. Thus, our formulation remains compatible with all scalable infrastructure for handling millions of Gaussians.
>
> *[1] Kerbl, Bernhard, et al. "A hierarchical 3d gaussian representation for real-time rendering of very large datasets." ACM Transactions on Graphics (TOG) 43.4 (2024): 1-15.*
>
> *[2] Liu, Yang, et al. "Citygaussian: Real-time high-quality large-scale scene rendering with gaussians." European Conference on Computer Vision. Cham: Springer Nature Switzerland, 2024.*
>
> *[3] Zhao, Hexu, et al. "On Scaling Up 3D Gaussian Splatting Training." The Thirteenth International Conference on Learning Representations.*
>
> ---
> ##### **Q3: If rendering quality keeps improving on pinhole datasets.**
>
> A: Thank you for the question.
>
> Yes, rendering quality on pinhole datasets continues to improve as the number of Gaussians increases, which is expected since additional primitives enhance representational capacity. However, the improvement becomes marginal once the scene geometry and appearance are sufficiently captured in limited FoV. This saturation behavior is shared across Gaussian-based renderers rather than being specific to our method.
>
> This observation is consistent with findings in prior literature—for example, Zhao et al. [1] show that when densifying from 5 to 9M Gaussians on MipNeRF scenes, reducing densification thresholds for further Gaussian growth yields marginal improvements, reinforcing that scaling Gaussian count past a certain point yields diminishing returns. The table below show the stats scaling up the Gaussian number on a MipNeRF pinhole scene (Bicycle):
>
> | PSNR ↑   | SSIM ↑ | # Gaussians  |
> |----------|--------|--------------|
> | 24.09    | 0.66   | 2185 K       |
> | 24.28    | 0.68   | 3036 K       |
> | 24.59    | 0.70   | 4155 K       |
> | 24.71    | 0.71   | 5273 K       |
> | 24.76    | 0.72   | 6579 K       |
> | 24.85    | 0.73   | 9396 K       |
>
> *[1] Zhao, Hexu, et al. "On Scaling Up 3D Gaussian Splatting Training." The Thirteenth International Conference on Learning Representations.*

---

> ### Author Response · Authors · 2025-11-23
> **Response (Part III)**
>
> ##### **Q4: Differences in transmittance and gradient computation between 3DGEER and GOF.**
>
> A:  Our closed-form expression and GOF [1]’s forward formulation arise from fundamentally different derivation assumptions. Specifically, ours is ***integral-preserving***, while GOF’s is ***amplitude-preserving*** (see [2]’s Figure 5 for these different formulations).
>
> Our derivation of the ray–Gaussian integral is based on the invariance of differential volume elements under geometric transformations, yielding a fully analytic integral. In contrast, GOF’s derivation ultimately reduces to a fuzzy-metaball [3] assumption, effectively using the maximum Gaussian density along the ray.
>
> The more substantial differences lie in the ***backward pathway*** and ***intermediate-variable caching***. We parameterize the canonical ray directly using cross-product identities, which enables gradients to propagate stably to the Gaussian parameters (scaling, rotation, and mean) through the original canonical-ray representation. GOF, however, rewrites the formulation into
>
> $
> D_{\mu,\Sigma}^{2} = a_{0} - \frac{a_{1}^{2}}{a_{2}},
> $ where [$a_0$, $a_1$, $a_2$] = [$\mathbf{p}_c^{\top}\Lambda\mathbf{p}_c$, $\mathbf{p}_c^{\top}\Lambda\mathbf{d}$, $\mathbf{d}^{\top}\Lambda\mathbf{d}$],
>
> and $\Lambda=W_{pca}^{\top}W_{pca}$ denoting the inverse covariance matrix,
> $\mathbf{p}_c$ the Gaussian center in view space, and $\mathbf{d}$ the ray direction in view space.
> The backward path is then constructed through this decomposition, resulting in gradient flow and numerical behavior that differ substantially from ours.
>
> More importantly, our gradient computation is tightly coupled with the ray–Gaussian association (PBF), because it determines which ray-bundled losses are eligible to backpropagate into each Gaussian—effectively acting as a spatial gating mechanism, whereas GOF adopts EWA. Similarly, the ray sampling induced by BEAP also influences gradient flow in training supervision.
>
> *[1] Yu, Zehao, Torsten Sattler, and Andreas Geiger. "Gaussian opacity fields: Efficient adaptive surface reconstruction in unbounded scenes." ACM Transactions on Graphics (ToG)43.6 (2024): 1-13.*
>
> *[2] Celarek, Adam, et al. "Does 3D Gaussian Splatting Need Accurate Volumetric Rendering?." Computer Graphics Forum. 2025.*
>
> *[3] Keselman, Leonid, and Martial Hebert. "Approximate differentiable rendering with algebraic surfaces." European Conference on Computer Vision. Cham: Springer Nature Switzerland, 2022.*

---

### Author Response · Authors · 2025-11-23
**General Response**

We sincerely thank all the reviewers for their constructive feedback and suggestions, which have helped us improve the quality and clarity of our work. Based on the reviewers' comments, we have revised our manuscript as follows:

***(All changes in manuscript are highlighted in blue for ease of reading and comparison)***

- Added the transmittance equation ablation experiments (*Section 4.4*, *Table 5* and *Figure 8*, full stats in *Table K.7*) based on the comments by reviewer **H5DA, uR7S, 62Nx**.
- Added BEAP on/off ablation experiments (*Section 4.5*, *Table 6*, *Figure K.5* and full stats in *Table K.8*) based on the comments by reviewer **H5DA, uR7S**.
- Corrected sentence ordering (sentence 2 and 3 were swapped in *Sec. 2.2*) based on the comments by reviewer **SpuR**.
- Corrected figures arrangement (*Figure K.2*) based on the comments by reviewer **62Nx**.
- Expanded the limitation discussion and future work discussion (*Appendix A*) based on the comments by reviewer **H5DA**.
- Expanded the discussion of prior works (*Appendix D.3*) based on the comments by reviewer **SpuR**.

We respond to each reviewer below to address the concerns. Let us know if further clarification or discussion is needed. Also, we will ensure that all discussions, tables and illustrations in the current revision are included in future versions.

---

### Meta-Review · Area_Chair_Bx49 · 2025-12-29

**Summary:**

This paper presents 3DGEER, a framework for projective-exact 3D Gaussian rendering under arbitrary camera models. The work addresses limitations in 3D Gaussian Splatting that arise from approximating 3D Gaussians as 2D projections, particularly for large field-of-view and fisheye cameras. The three main contributions are: (1) closed-form ray-Gaussian integration via canonical space transformation; (2) Particle Bounding Frustum (PBF) for efficient ray-Gaussian association without BVH; and (3) Bipolar Equiangular Projection (BEAP) for balanced angular sampling.

All four reviewers scored the paper at or above acceptance threshold (`H5DA`: 6, `uR7S`: 6, `SpuR`: 6, `62Nx`: 10). The paper demonstrates 5× speedup over ray-based baselines and 1.6-2.4× fewer Gaussians than approximation methods. Reviewers praised the theoretical foundation, efficiency, generalizability, and experimental comprehensiveness. Initial concerns about missing ablations and limited prior work discussion were addressed through revisions (Sections 4.4-4.5, Appendix D.3). However, Reviewer `SpuR` maintains concerns about how the work is positioned relative to concurrent methods (HTGS, 2DGS).

**Reviewer Concerns:**

**Addressed concerns**: Reviewers `H5DA` and `uR7S` requested ablation studies to isolate component contributions; authors responded with comprehensive experiments in Sections 4.4-4.5 demonstrating that projective exactness reduces Gaussian count by 1.6-2.4× and BEAP outperforms alternatives. `H5DA` requested limitations discussion; authors added Appendix A on ordering/self-occlusion tradeoffs. Reviewer `62Nx` identified figure errors that were corrected. Various technical clarifications were provided regarding Gaussian reduction mechanisms, scalability, and differences from related methods (GOF). Reviewers have not yet confirmed satisfaction with these additions.

**Outstanding concern**: Reviewer `SpuR` raised follow-up concerns about positioning relative to concurrent work (HTGS, 2DGS), stating "currently uncertain about acceptance." `SpuR` questions whether the relationship to prior work should be framed as a "generalization" versus "alternative formulation," suggests the title "Generic Cameras" may overstate the contribution, and requests more prominent main-text comparison. Authors responded by changing terminology from "special case" to "pinhole alternative," adding HTGS to Table 1, defending "generic cameras" as standard terminology in vision literature (Mei & Rives 2007; Kannala & Brandt 2006), and noting prior work was cited in the original submission. The disagreement centers on presentation framing rather than technical validity, which remains undisputed.

**Reviewer Scores:**

**Current Scores:**
- **Reviewer `H5DA`**: 6 (marginally above threshold) - likely to remain at 6, possibly increase to 7 given satisfactory ablation responses
- **Reviewer `uR7S`**: 6 (marginally above threshold) - likely to remain at 6, possibly increase to 7 given comprehensive ablation responses
- **Reviewer `SpuR`**: 6 (marginally above threshold) - *uncertain*, could potentially decrease to 5 if positioning concerns remain unresolved, or maintain 6 if satisfied with revisions
- **Reviewer `62Nx`**: 10 (strong accept, oral/spotlight) - likely to remain at 10, endorsement unchanged

**Expected post-discussion scores**: 6, 6-7, 5-6, 10 (median: 6-7)

---

### Decision · Program_Chairs · 2026-01-26

Accept (Poster)